# CaV1 and CaV2 calcium channels mediate the release of distinct pools of synaptic vesicles

Brian D Mueller[1†], Sean A Merrill[1†], Shigeki Watanabe[1], Ping Liu[2], Longgang Niu[2], Anish Singh[1], Pablo Maldonado-Catala[3], Alex Cherry[1], Matthew S Rich[1], Malan Silva[1], Andres Villu Maricq[3], Zhao-Wen Wang[2], Erik M Jorgensen[1]*

[1]Howard Hughes Medical Institute, School of Biological Sciences, University of Utah, Salt Lake City, United States; [2]Department of Neuroscience, University of Connecticut Medical School, Farmington, United States; [3]Department of Neurobiology, University of Utah, Salt Lake City, United States

**\*For correspondence:** jorgensen@bioscience.utah.edu

[†]These authors contributed equally to this work

**Competing interest:** The authors declare that no competing interests exist.

**Abstract** Activation of voltage-gated calcium channels at presynaptic terminals leads to local increases in calcium and the fusion of synaptic vesicles containing neurotransmitter. Presynaptic output is a function of the density of calcium channels, the dynamic properties of the channel, the distance to docked vesicles, and the release probability at the docking site. We demonstrate that at *Caenorhabditis elegans* neuromuscular junctions two different classes of voltage-gated calcium channels, CaV2 and CaV1, mediate the release of distinct pools of synaptic vesicles. CaV2 channels are concentrated in densely packed clusters ~250 nm in diameter with the active zone proteins Neurexin, α-Liprin, SYDE, ELKS/CAST, RIM-BP, α-Catulin, and MAGI1. CaV2 channels are colocalized with the priming protein UNC-13L and mediate the fusion of vesicles docked within 33 nm of the dense projection. CaV2 activity is amplified by ryanodine receptor release of calcium from internal stores, triggering fusion up to 165 nm from the dense projection. By contrast, CaV1 channels are dispersed in the synaptic varicosity, and are colocalized with UNC-13S. CaV1 and ryanodine receptors are separated by just 40 nm, and vesicle fusion mediated by CaV1 is completely dependent on the ryanodine receptor. Distinct synaptic vesicle pools, released by different calcium channels, could be used to tune the speed, voltage-dependence, and quantal content of neurotransmitter release.

## Editor's evaluation

Using an elegant combination of cutting-edge techniques, the authors show that in the neuromuscular junction of the nematode *C. elegans* two different classes of voltage-activated calcium channels differentially trigger exocytosis of distinct pools of synaptic vesicles, one docked to the active zone and a second one localized more distant from the active zone. These findings provide fascinating and novel insights into a classical problem of presynaptic physiology.

## Introduction

Synaptic vesicles fuse to the plasma membrane within the presynaptic bouton in a domain called the active zone, and the intricate molecular architecture within the active zone determines the dynamics of the neurotransmitter release (*Guzikowski and Kavalali, 2021*). Vesicle fusion is driven by calcium influx and binding to the calcium sensor synaptotagmin on the synaptic vesicle (*Geppert et al., 1994*; *Littleton et al., 1993*; *Ward et al., 2004*). The coupling of calcium channels to fusion sites determines the transfer function of synapses to depolarizing inputs (*Eggermann et al., 2011*; *Eguchi et al., 2022*;

*Özçete and Moser, 2021*; *Rebola et al., 2019*), and thus synaptic activity depends on three features of calcium-mediated vesicular fusion: the dynamic properties of the calcium channel, the concentration of calcium at the fusion site, and the release probability of the vesicle. These features are dictated by calcium channel type and location, by the distance to docked vesicles, and by the activity of the priming protein Unc13. Here, we characterize these features at the *Caenorhabditis elegans* neuromuscular junction.

Voltage-gated calcium channels are divided into three molecular families: CaV1, CaV2, and CaV3, each with fundamentally different dynamic properties, including voltage-sensitive activation and inactivation (*Catterall et al., 2005*; *Nowycky et al., 1985*). Each of these channel classes is primarily associated with tissue-specific functions: In muscle, CaV1 (L-type) channels mediate contraction and are coupled to the ryanodine receptor to release internal calcium stores (RyR). In neurons, CaV2 (P/Q, N, and R-type) channels drive synaptic transmission. In neurons and excitable cells, CaV3 (T-type) regulate action potential oscillations and pacemaker frequencies (*Dolphin, 2021*) These tissue-specific roles are not exclusive, for example, the CaV1 variants CaV1.3 and CaV1.4 are associated with neurotransmitter release in hair cells and photoreceptors, respectively (*Dolphin and Lee, 2020*).

In the nematode *C. elegans*, each class is represented by a single gene: CaV1 (*egl-19*), CaV2 (*unc-2*), CaV3 (*cca-1*), and RyR (*unc-68*). In all animals, CaV2 is the main calcium channel for synaptic transmission (*Richmond et al., 2001*; *Smith et al., 1996*; *Tsien et al., 1988*; *Tsien and Tsien, 1990*; *Zheng et al., 1995*). Unlike other animals, nematodes lack voltage-gated sodium channels and neurotransmission is typically mediated via graded release, however some interneurons use action potentials (*Liu et al., 2013*; *Liu et al., 2009*; *Mellem et al., 2008*). In *unc-2* mutants, which lack the CaV2 channel, the frequency of tonic miniature currents ('minis') is severely reduced; however, some release remains (*Richmond et al., 2001*; *Tong et al., 2017*). Physiological studies suggest CaV1 can also contribute to neurotransmission; CaV1 channel blockers reduce tonic minis *by half* (*Tong et al., 2017*). However, the role of CaV1 channels at synapses in *C. elegans* is complicated because CaV1 also contributes to calcium-mediated action potentials in neurons and is required in the body muscle for viability (*Lee et al., 1997*; *Liu et al., 2018a*). Finally, the ryanodine receptor also contributes to neurotransmission, and is specifically required for multivesicular release (*Chen et al., 2017*; *Liu et al., 2005*).

The distance between the calcium channel and docked and primed vesicle must be very short. After the channels close, diffusion causes a rapid drop in the concentration of calcium at sites of vesicle fusion (*Dittman and Ryan, 2019*). Free calcium is further reduced by calcium buffers and calcium pumps (*Blaustein, 1988*; *Eggermann et al., 2011*). Because intracellular calcium is extremely low (0.05 μM), and levels required for fusion are relatively high (half-maximal 10 μM) (*Courtney et al., 2018*; *Schneggenburger and Neher, 2000*), the effective range of calcium around a single voltage-gated calcium channel is predicted to be only 20 nm for evoked fusion, a 'nanodomain' not much larger than the diameter of the calcium channel itself (*Fedchyshyn and Wang, 2005*; *Weber et al., 2010*). The synaptic vesicle in vertebrates is 45 nm in diameter; in *C. elegans* vesicles are somewhat smaller, 32 nm in diameter; nevertheless, these data suggest that the vesicle must be nearly on top of the calcium channel.

The presence of vesicles docked at release sites and the probability of vesicle fusion depends on the active zone protein Unc13 (*Dittman, 2019*; *Neher and Brose, 2018*). Unc13 tethers vesicles to the active zone membrane through C2B and C2C domains which flank the MUN domain (*Imig et al., 2014*; *Quade et al., 2019*). The central MUN domain interacts with the SNARE protein syntaxin (*Augustin et al., 1999*; *Lai et al., 2017*; *Yang et al., 2015*) and promotes the open state of syntaxin to initiate SNARE pairing (*Richmond et al., 2001*). Moreover, binding of DAG and calcium to Unc13 regulates the differential release probabilities of primed vesicles (*Basu et al., 2007*; *Michelassi et al., 2017*; *Neher and Brose, 2018*). In the absence of Unc13, synaptic vesicles fail to dock at release sites (*Hammarlund et al., 2007*; *Imig et al., 2014*; *Richmond et al., 1999*; *Siksou et al., 2009*).

Here, we demonstrate in *C. elegans* that two different classes of voltage-gated calcium channels, CaV2 (UNC-2) and CaV1 (EGL-19) mediate the release of two physiologically distinct pools of synaptic vesicles as described in a previous study (*Tong et al., 2017*). We also show that a third calcium channel, the ryanodine receptor (RyR / UNC-68), is essential for CaV1-mediated vesicle release. Time-resolved electron microscopy in calcium channel mutants demonstrates that these channels mediate fusion of spatially distinct pools of synaptic vesicles in the same synaptic varicosity. Finally, we use super-resolution fluorescence microscopy to demonstrate that CaV2 is localized with UNC-13L at

the dense projection, and that CaV1 and RyR colocalize with UNC-13S at distal sites. Altogether, we describe two pools of synaptic vesicles: (1) The central pool is localized adjacent to the dense projection, vesicles are docked by UNC-13L, and released by a dense cluster of CaV2 channels. (2) The lateral pool of vesicles is broadly distributed, docked by UNC-13S, and released by dispersed CaV1 and RyR channels.

## Results

### CaV1 and CaV2 calcium channels have partially overlapping functions

The genome of *C. elegans* contains only a single gene for each major voltage-gated calcium channel class: CaV1 (*egl-19*), CaV2 (*unc-2*), CaV3 (*cca-1*), and a single calcium-gated RyR (*unc-68*) (hereafter, referred to by their common names). Loss of the CaV3 T-type channel does not affect neurotransmitter release in acetylcholine neurons (*Liu et al., 2018b*). However, loss of any other calcium channel results in impaired neurotransmission (*Liu et al., 2005*; *Richmond et al., 2001*; *Tong et al., 2017*). Null mutants lacking either CaV2 (*unc-2(lj1)*) or RyR (*unc-68(e540)*) are viable. *unc-2(lj1)* is a large deletion and frame shift, and *unc-68(e540)* is a G>A splice acceptor mutation near the middle of the protein, likely causing a null phenotype (*Sakube et al., 1997*; this paper). CaV1 null mutants (*egl-19(st556)*) die as embryos due to a loss of muscle function during morphogenesis (*Lee et al., 1997*). We rescued the CaV1 null mutant using a muscle promoter expressed early in development; since this strain lacks CaV1 in the nervous system, we refer to it as 'CaV1(Δns)'.

To determine whether these channels function cooperatively or in parallel, we tested for synthetic interactions between mutations of these channel types. The double mutant CaV1(Δns) RyR(-) is viable, and is no worse than the RyR null, consistent with their coupled function (*Figure 1A*). However, CaV1(Δns) CaV2(-) double mutants and RyR(-) CaV2(-) double mutants exhibit a synthetic lethal interaction. These data suggest that calcium influx from CaV1-RyR acts redundantly, and in parallel, with CaV2 to sustain neuronal function essential for viability.

CaV1(Δns) animals are uncoordinated, and the phenotypes are fully rescued by the expression of CaV1 in the nervous system (*Figure 1B*). To determine the role of CaV1 on locomotion, we characterized animal movement on agar plates using worm tracker software. CaV1(Δns) worms moved slower than wild-type worms, and consequently travelled shorter distances during the observation period. Additionally, CaV1(Δns) worms reversed more frequently, had shorter durations of forward movement, and moved forward shorter distances than wild type (*Figure 1C–G*). The distance travelled while backing tended to be shorter in the CaV1(Δns) animals; whereas the rescued animals travelled longer distances in reverse, about a full body length (*Figure 1H*). Reversals in all genotypes were similar in duration (*Figure 1I*). These results indicate that the speed of locomotion is dependent on CaV1 function in the nervous system, and that CaV1 also biases the bistable locomotory circuit toward forward locomotion (*Zheng et al., 1999*).

If CaV1 and RyR are functioning in the same pathway then worms lacking expression of RyR in the nervous system should phenotypically mimic CaV1(Δns). *unc-68(syb216)* animals lack the neuronal-specific isoform of RyR, but express ~10% levels of RyR in neurons from the muscle promoter (*Ma et al., 2016*; *Marques et al., 2020*). We will refer to this strain as RyR(Δns) for simplicity. Like CaV1(Δns) animals, RyR(Δns) animals are significantly slower than the wild type; however, they do not exhibit the same increase in reversal frequency (*Figure 1—figure supplement 1A–I*).

### CaV2 and CaV1 regulate distinct pools of synaptic vesicles

To determine if CaV1 is playing a direct role in synaptic transmission, we recorded spontaneous synaptic currents at neuromuscular junctions. Body muscles were voltage-clamped at a holding potential of –60 mV, and miniature postsynaptic currents recorded under high chloride internal pipette conditions, in which GABA and ACh release generates inward currents. Miniature postsynaptic currents ('minis') are caused by the release of neurotransmitter from one or a few synaptic vesicles. In the nematode, vesicle fusion is graded, that is, it is proportional to membrane depolarization (*Liu et al., 2014*; *Liu et al., 2009*); high frequency of these 'tonic' minis drives calcium action potentials in the muscles (*Liu et al., 2011*).

The rate of miniature postsynaptic currents (minis/s) compared to the wild type (32.2+/-2.1 minis/s) was significantly reduced in each of the single mutants, CaV2 (18.7+/-3.5 minis/s), CaV1(Δns)

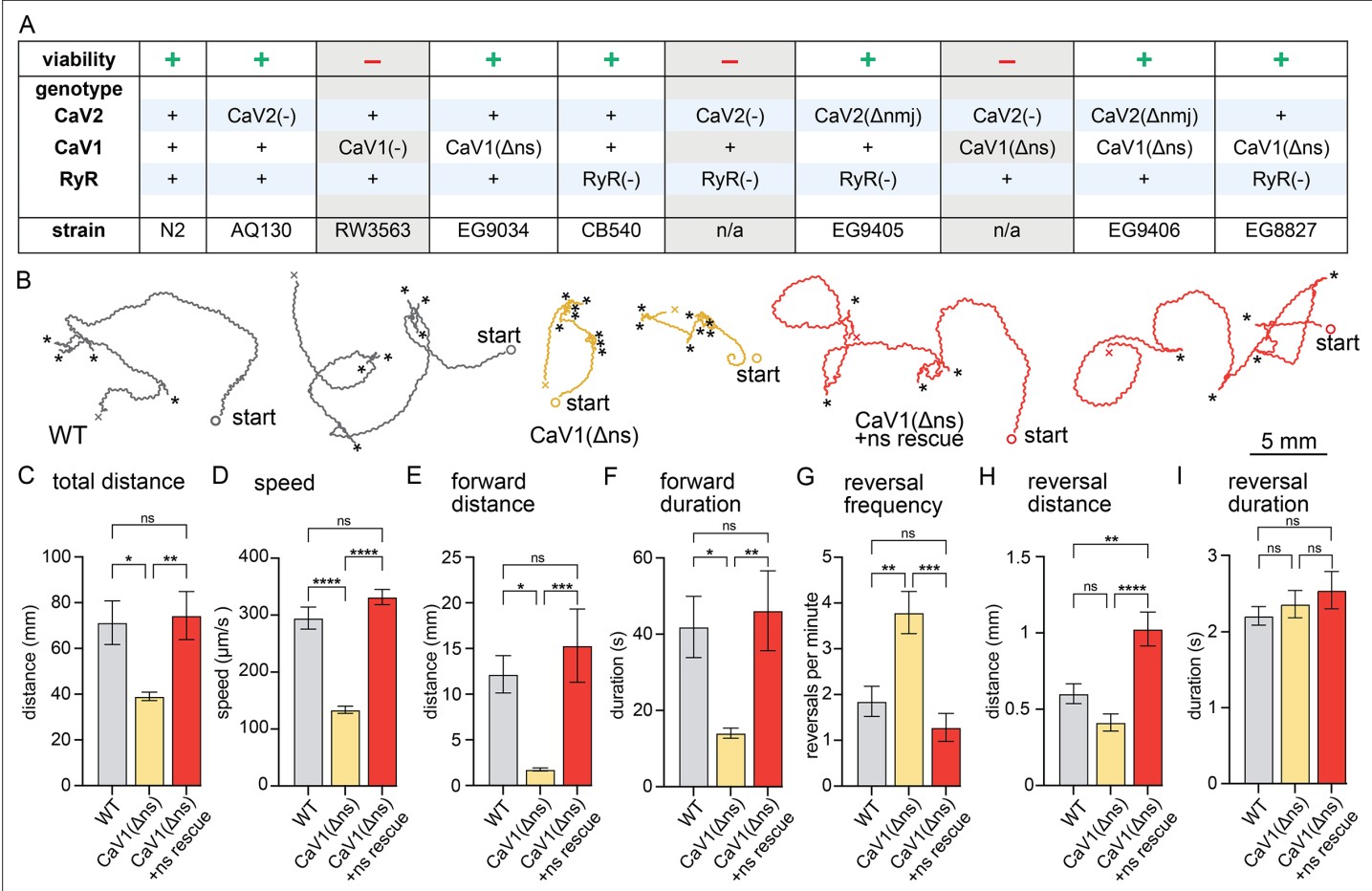

**Figure 1.** nervous system CaV1 is required for normal behavior and locomotion. (**A**) Viability of calcium channel double mutants. (**B**) Worm Tracks. Healthy animals were tracked for 5 minutes with a frame rate of 8 frames per second. The path the animal created was plotted, starting point is indicated, asterisks represent reversal events. (**C**) Total average distance animals travelled per animal during the 5 minute interval by genotype. Wild-type 71.2±9.5 mm. CaV1(Δns) 39.0±1.8 mm. CaV1(Δns)+rescue 74.4±10.5 mm. (**D**) Average speed, including both forward and backward bouts but excluding pauses, for the duration of the assay. Wild-type 294.9 µm/s±19.4 µm/s. CaV1(Δns) 133.8 µm/s±6.3 µm/s. CaV1(Δns)+rescue 331.7 µm/s±13.3 µm/s. (**E**) Average distance of forward locomotion between reversal events that animals travelled by genotype. Wild-type 12.2 mm ±2.0 mm. CaV1(Δns) 17.7 mm ±1.7 mm. CaV1(Δns)+rescue 15.3 mm ±4.0 mm. (**F**) Average duration of forward run between reversal events. Wild-type 41.9±8 s. CaV1(Δns) 14.1±1.3 s. CaV1(Δns)+rescue 46.2±10.5 s. (**G**) Average number of reversal events per minute exhibited by animals by genotype. Wild-type 1.9±0.3 events. CaV1(Δns) 3.8±0.5 events. CaV1(Δns)+rescue 1.3±0.3 events. (**H**) Average distance travelled in reverse per animal by genotype. Wild-type 601.9±65.1 µm. CaV1(Δns) 413.2±56.8 µm. CaV1(Δns)+rescue 1026±111.1 µm. (**I**) Average duration of reversal run. Wild-type 2.2 +- 0.1 s. CaV1(Δns) 2.4+/-0.2 s. CaV1(Δns)+rescue 2.5 +- 0.2 s. Wild-type n=11, CaV1(Δns) n=16, CaV1(Δns)+rescue n=13. Error bars reported in SEM. Genotypes were blinded. One-way ANOVA with Tukey's multiple comparisons was used to calculate p-value. *p<0.05, **p<0.005, ***p<0.001, ****p<0.0005. Data available as *Figure 1—source data 1*.

The online version of this article includes the following source data and figure supplement(s) for figure 1:

**Source data 1.** Worm behaviorial data.

**Source data 2.** Table of calcium channel mutant viability, genotype, and strain designation.

**Figure supplement 1.** RyR neuronal isoform knockout moves slower than wildtype.

(18.7+/-3.6 minis/s), and RyR (20.9+/-1.5 minis/s) (*Figure 2A and B*). Because CaV2 CaV1(Δns) double mutants are synthetic lethal, we acutely blocked CaV1 using 10 µM nemadipine (*Kwok et al., 2006*). Nemadipine reduced minis in the wild type (+nema 18.4+/-2 minis/s) to a similar level as the CaV1(Δns) mutant alone, and did not further reduce mini frequency when paired with the CaV1(Δns) mutant (+nema 19.1+/-2.7 mini/s). These data demonstrate that nemadipine is an effective blocker of CaV1 and does not block CaV2.

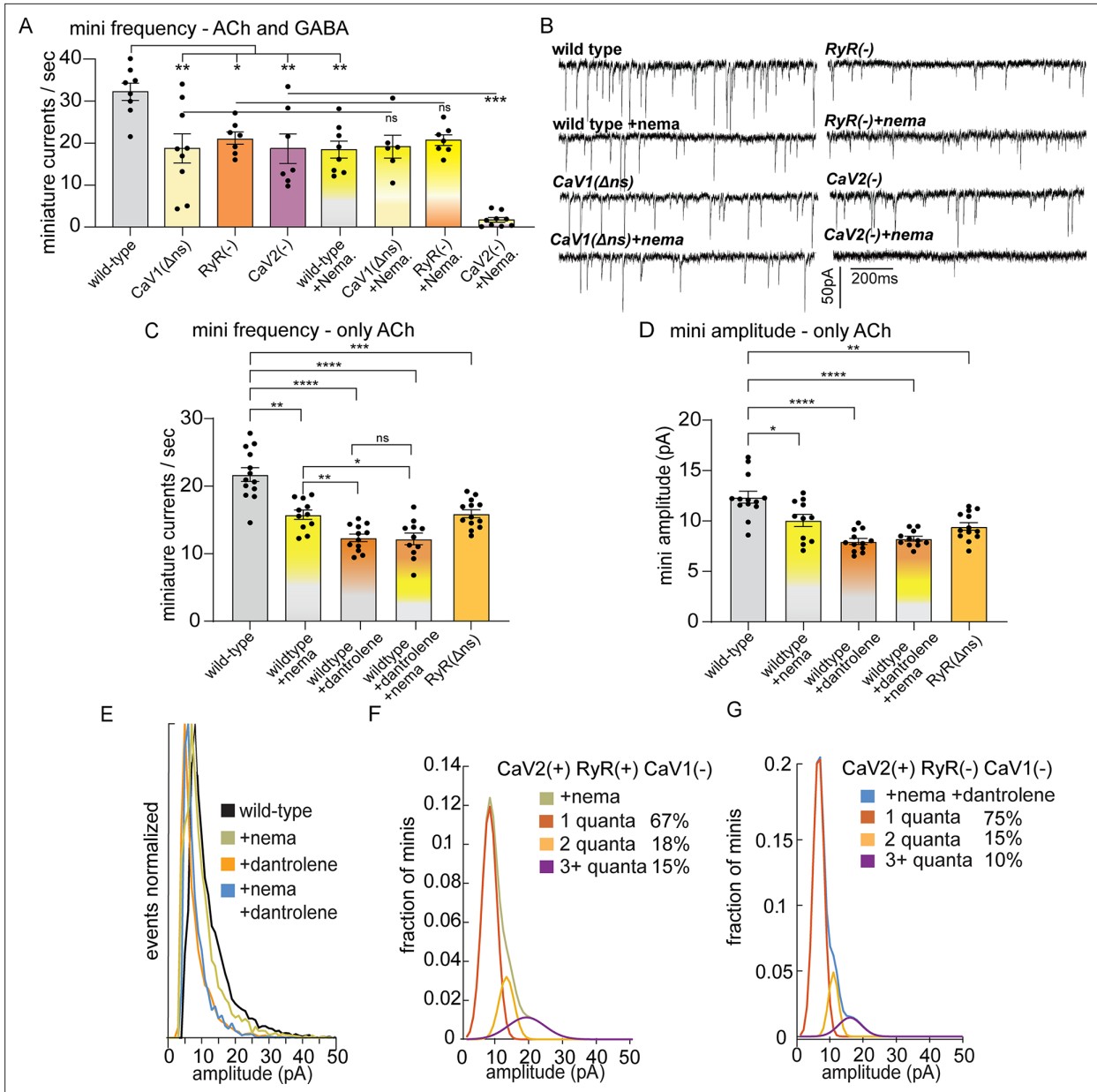

**Figure 2.** CaV2 and CaV1-RyR fuse different vesicle pools. (**A**) Spontaneous miniature currents mediated by CaV1 and RyR are inhibited by nemadipine. Wild-type: 32.2±2.1 minis/s n=8, wild type with nemadipine (10 µM): 18.4±2.1 mini/s n=8. CaV1(Δns): 18.7±3.1 minis/s n=9, with nemadipine 19.1±2.7 mini/s n=6. RyR(-): 20.9±1.5 minis/s n=7, with nemadipine 20.7±1.3 mini/s n=7. CaV2(-): 18.7±3.5 minis/s n=7, with nemadipine 1.7±0.6 mini/s n=9. One-way ANOVA with Dunnett's multiple comparisons test and one-way ANOVA with Tukey's multiple comparisons tests were used to calculate significance. GABA and ACh release generated inward currents in this preparation. (**B**) Sample traces of spontaneous release in 0.5 mM extracellular calcium. GABA and ACh release generated inward currents in this preparation. (**C**) Spontaneous miniature currents from acetylcholine release are reduced with pharmacological block of CaV1 and RyR. Wild-type: 21.7+/-1 mini/s n=13. Wild-type +nemadipine (10 µm): 15.8+/-0.7 mini/s n=11. Wild-type +dantrolene: 12.36+/-0.6 mini/s n=12. Wild-type +nemadipine and dantrolene: 12.2+/-0.9 mini/s n=11. RyR(ns-): 15.9+/-0.6 mini/s n=13. Brown-Forsythe and Welch ANOVA with T3 Dunnett's multiple comparisons test were used to calculate significance. (**D**) Dantrolene reduces miniature current amplitude from acetylcholine release. Wild-type: 12.4+/-0.6 pA n=13. Wild-type +nemadipine (10 µm): 10.1+/-0.6 pA n=11. Wild-type +dantrolene: 8.0+/-0.3 pA n=12. Wild-type +nemadipine and dantrolene: 8.2+/-0.24 pA n=11. RyR(ns-): 9.5+/-0.4 pA n=13. Brown-Forsythe and Welch ANOVA with T3 Dunnett's multiple comparisons test were used to calculate significance. (**E**) Frequency distribution of mini amplitudes from acetylcholine release in wild-type and with pharmacological block of CaV1 or RyR, normalized to mode. (**F**) Quantal analysis of post-synaptic amplitudes from wild-type animals treated with nemadipine. Mini amplitudes were transformed into 1 pA bins. The wild-type +nema distribution of amplitudes was fit with a three-term Gaussian convolution to isolate 1, 2 and 3 quantal events, each centered around the mode: 7±1 pa, 14±2 pa and 21±3 pa (khaki). 1-quanta (rust) accounted for 67% of fusions. 2-quanta (butterscotch) 18%. 3-quanta (violet) 15%. (**G**) Quantal analysis of post-synaptic amplitudes from wild-type

*Figure 2 continued on next page*

*Figure 2 continued*

animals treated with dantrolene and nemadipine. Mini amplitudes were transformed into 1 pA bins. The wild-type +nema distribution of amplitudes was fit with a three-term Gaussian convolution to isolate 1, 2, and 3 quantal events, each centered around the mode: 6±1 pa, 12±2 pa and 18±3 pa (blue). (blue). 1-quanta (rust) accounted for 75% of fusions. 2-quanta (butterscotch) 15%. 3-quanta (violet) 10%. For all recordings, Vm = –60 mV, 0.5 mM calcium. Error bars reported in SEM. *p<0.05, **p<0.005, ***p<0.001, ****p<0.0005. Data available as *Figure 2—source data 1*.

The online version of this article includes the following source data and figure supplement(s) for figure 2:

**Source data 1.** Electrophysiology data from calcium channel mutants with pharmacological block of calcium channels.

**Figure supplement 1.** The ryanodine receptor acts in parallel to CaV2.

**Figure supplement 1—source data 1.** Electrophysiology data from calcium channel double mutants.

Nemadipine application in the RyR mutant did not exacerbate the phenotype ('RyR(-)+nema', 20.7+/-1.3 mini/s), indicating that CaV1 and RyR function is coupled at neuromuscular junctions. To determine if the plasma membrane channels CaV1 and CaV2 are required together for all neurotransmitter release, we blocked CaV1 in the CaV2 null mutant. Application of nemadipine almost completely abolished mini frequency in the CaV2 mutant (+nema 1.7+/-0.6 mini/s). We conclude that all vesicle fusion at neuromuscular junctions relies on CaV1 and CaV2, each contributing about half of the minis, in agreement with an earlier study (*Tong et al., 2017*). In addition, calcium influx through CaV1 is not sufficient for vesicle fusion; CaV1 relies on internal calcium stores released by RyR to fuse synaptic vesicles.

We further tested the roles of calcium channels using double mutants. We found that the rate of minis in the CaV1(Δns) RyR double mutant (19.6+/-2 mini/s) was similar to the single CaV1(Δns) and RyR mutants, supporting the hypothesis that CaV1 relies on coupling to RyR to activate neurotransmitter release. Because CaV1 CaV2 double mutants exhibit synthetic lethality (*Figure 1A*), we generated a mosaic CaV2 strain in which the channel was expressed in acetylcholine head neurons using a tissue-specific promoter ('P*unc-17h*') in the CaV2 null mutant, referred to as 'CaV2(Δnmj)' (*Hammarlund et al., 2007*; *Topalidou et al., 2016*). Expression of CaV2 in head neurons bypassed the synthetic lethality of both the CaV1(Δns) CaV2(-) double mutant, and the RyR(-) CaV2(-) double mutant, but the rescued animals exhibit a synthetic paralyzed phenotype. The mini rates of CaV2(Δnmj) CaV1(Δns) double mutants (12.8+/-2.4 mini/s) and CaV2(Δnmj) RyR double mutants (11.3+/-2.1 mini/s) were significantly reduced, but not completely abolished (*Figure 2—figure supplement 1A*). The remaining neuronal activity in these strains is likely due to CaV2 expression in the sublateral cord motor neurons located in the head and extend long processes that synapse onto body muscles. In summary, these results demonstrate that CaV1 and RyR are interdependent, and act in parallel to CaV2.

To confirm that the reduction of minis observed in the *unc-68* null mutant was due to an acute loss of RyR function rather than a developmental defect, we blocked RyR by applying dantrolene (10 μM), a specific RyR inhibitor (*Song et al., 1993*; *Xu et al., 2001*), either alone or in combination with the CaV1 blocker nemadipine. The recordings were performed using a low-chloride pipette solution at a holding voltage equal to the chloride equilibrium potential so that only minis mediated by acetylcholine were detected. The frequency of minis was reduced in all treatments compared to the wild type (*Figure 2C*; wild type 21.7±1.0 mini/s; dantrolene alone 12.36+/-0.6 mini/s; nemadipine alone 15.8+/-0.7 mini/s; dantrolene plus nemadipine 12.2+/-0.9 mini/s). Furthermore, nemadipine did not exacerbate the inhibitory effect of dantrolene on minis. Again, CaV1 is reliant on RyR for neurotransmission.

To confirm that RyR is acting presynaptically we analyzed strains rescued in neurons or in muscle. Previously, we rescued minis in a null mutant by expressing wild-type *unc-68* in neurons but not muscle cells, suggesting that RyRs regulate minis by acting presynaptically (*Liu et al., 2005*). Expression from *unc-68* is driven by an upstream muscle promoter and a downstream neuronal promoter (*Chen et al., 2017*; *Marques et al., 2020*). To confirm the presynaptic function of RyRs, we recorded minis from the RyR(Δns) mutant *unc-68(syb216)* (*Figure 2C*). The frequency of minis was significantly reduced compared to the wild type (wild type 21.7±1.0 mini/s; RyR(Δns) 15.9±0.6 mini/s; p=0.001). The reduction in minis in RyR(Δns) supports the conclusion that RyR is required presynaptically for normal levels of synaptic vesicle fusion. Mini frequency in RyR(Δns) was slightly higher than pharmacological block by dantrolene (12.36+/-0.6 mini/s; p=0.002). This result is consistent with the observation ~10% of RyR transcripts in neurons are expressed from the 'muscle' promoter in this strain (*Marques et al., 2020*).

## RyR is required for multiquantal release

The amplitude of a miniature currents recorded from the muscle reflects the amount of neurotransmitter released by the synaptic bouton. The mode of mini amplitudes represents miniature currents from single vesicle fusions – single fusions are the most probable event. Using recording conditions in which only acetylcholine currents are detected, the modal value of miniature currents was similar in all strains (wild type 8 pA, CaV1 block nemadipine 7 pA, RyR block dantrolene 5 pA, CaV1 +RyR block nemadipine +dantrolene 6 pA, RyR(Δns) 6 pA), suggesting that the receptor field is similar for most single vesicle fusions. Simultaneous fusion of multiple vesicles – multiquantal release – will increase the mean current amplitude. The mean current amplitude from acetylcholine release was significantly reduced by pharmacological or genetic block of RyR in neurons (wild type:12.4±0.6 pA; dantrolene: 8.0±0.3 pA; RyR(Δns): 9.5±0.4 pA) (*Figure 2D and E*). The presence of large current events was not reduced by mutation of CaV2 (*Figure 2—figure supplement 1D*, G). Block of CaV1 by nemadipine caused a decrease in the mean current amplitude (wild type:12.4±0.6 pA; nemadipine: 10.1±0.6 pA; nemadipine +dantrolene: 8.2±0.24 pA), suggesting that CaV1 contributes to multiquantal release.

Similar results were observed in mutants lacking these channels, using recording conditions in which both acetylcholine and GABA release were detected. The modal value was similar in all genotypes (WT 10 pA, CaV1Δns 10 pA, RyR 11 pA, CaV2 8 pA, WT +nema 10 pA, CaV1Δns +nema 8 pA, RyR +nema 10 pA, CaV2 +nema 11 pA). However, the mean amplitude of miniature currents was reduced in the RyR(-) mutant (15.5±0.6 pA) compared to the wild type (22.3±1.9 pA; *Figure 2—figure supplement 1B–G*). Together, these data indicate that CaV1 is coupled to the ryanodine receptor at synapses to drive multiquantal release.

CaV2 is also functionally coupled to RyR to drive multiquantal release in acetylcholine motor neurons. When CaV1 channels are blocked by nemadipine, the remaining miniature currents rely on CaV2 and RyR (*Figure 2D*), and mini amplitudes arising solely from CaV2 were reduced by further blocking RyR (nemadipine: 10.1+/-0.6 pA; dantrolene +nemadipine: 8.2+/-0.24 pA), suggesting that RyR also responds to calcium from CaV2. To estimate individual contributions to multiquantal release, we fit the distribution of current amplitudes assuming simultaneous fusions from multiple single vesicles (*Figure 2E*). In the presence of CaV2 and RyR 34% of currents are multiquantal; blocking RyR reduces multiquantal release to 25% (*Figure 2F and G*). These fits indicate that RyR contributes to CaV2-mediated vesicle fusion by enhancing multiquantal events, though multiquantal events can also be attributed to CaV2.

Together, these data demonstrate that CaV2 and CaV1 channels regulate the release of separate synaptic vesicle pools at neuromuscular junctions. CaV1 requires the ryanodine receptor for any vesicle fusion, consistent with the known relationship of these channels in calcium-activated calcium release. Calcium influx through CaV2 is sufficient to fuse vesicles on its own, although neurotransmitter release is amplified by calcium release from internal stores via the ryanodine receptor.

## CaV2 and CaV1 mediate fusion of separate vesicle pools at single synapses

The physiology data suggest that CaV2 and CaV1 mediate the release of distinct synaptic vesicle pools at neuromuscular junctions. To determine whether these calcium channels regulate spatially distinct pools at the same synaptic varicosity, time-resolved 'flash-and-freeze' electron microscopy was used to characterize fusing vesicle pools (*Watanabe et al., 2013*). Transgenic animals expressing channelrhodopsin (ChIEF) in acetylcholine neurons were loaded into a high-pressure freezing chamber and stimulated with a 20 ms light pulse to depolarize neurons and activate synaptic calcium channels. Animals were frozen 50ms after stimulation; control animals were treated identically but not stimulated. Frozen samples were fixed by freeze substitution, embedded in plastic, and sectioned for electron microscopy (*Watanabe et al., 2013*). Docked vesicles were defined as those in contact with the plasma membrane; docking was scored blind to treatment and genotype (*Figure 3A and B*). The distance from the dense projection to the docked vesicle was plotted on the X-axis (*Figure 3C*). Decreases in docked vesicles after stimulation were assumed to be the result of synaptic vesicle fusion, although calcium influx could cause some vesicles to undock and return to the cytoplasm (*Kusick et al., 2020*).

To identify vesicle fusions associated with particular calcium channels, we analyzed the distribution of docked vesicles in mutant animals. In unstimulated animals, docked vesicles were clustered around

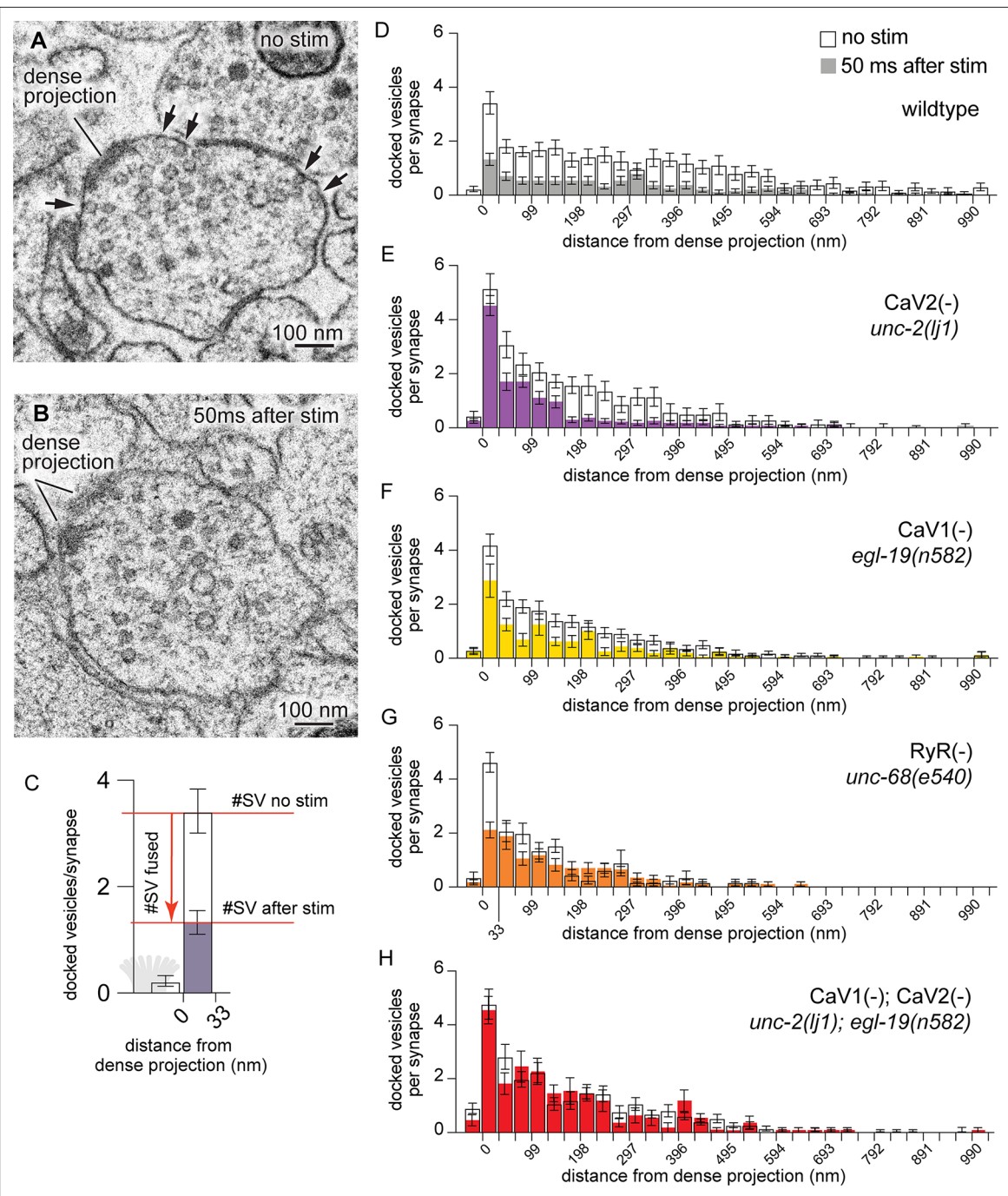

**Figure 3.** CaV2 and CaV1-RyR act at distinct vesicle release sites. (**A**) Docked vesicles (black arrows) are present at synapses in electron micrographs of unstimulated animals. (**B**) Docked vesicles are reduced 50ms after channelrhodopsin stimulation. (**C**) Vesicle fusion. The number of synaptic vesicles that fuse can be calculated as the number of docked vesicles lost by stimulation. Dense projection indicated in gray in the 0 nm bin. (**D–H**) Average number of docked vesicles per synapse at a given distance from the dense projection with, or without, light stimulation of channelrhodopsin. (**D**) Wild-type animals exhibit fewer docked vesicles at all locations after stimulation. Wild type (no stimulation), n=26 synapses. Wild type (stimulated) n=24 synapses. (**E**) In the CaV2 null mutant *unc-2(lj1)*, vesicles fuse greater than 33 nm from the dense projection; vesicle fusion is reduced directly adjacent to the dense projection. No stimulation n=14, stimulated n=27 synapses. (**F**) The CaV1 hypomorphic mutant *egl-19(n582)* exhibits reduced fusions at all distances. No stimulation n=29 synapses, stimulated n=16 synapses. (**G**) The RyR mutant *unc-68(e540)* exhibits fusions adjacent to the dense projection, but lacks fusions of lateral vesicles. No stimulation n=11 synapses, stimulated n=17 synapses. (**H**) The CaV1 CaV2 double mutant, *egl-19(n582) unc-2(lj1)*, lacks fusion of all docked vesicles after stimulation. No stimulation n=24 synapses, stimulated n=17 synapses. Micrographs were segmented blind to treatment and genotype. Bin size was fixed at 33 nm to be consistent with our section thickness in case 3D reconstruction of synapses is required. Error bars SEM, N=2 animals for each condition. Data available as *Figure 3—source data 1*.

*Figure 3 continued on next page*

*Figure 3 continued*

The online version of this article includes the following source data and figure supplement(s) for figure 3:

**Source data 1.** Quantification of the number of docked vesicles in calcium channel mutants.

**Figure supplement 1.** Statistical analysis of synaptic vesicle fusion in domains of the active zone.

dense projections, although many were observed at lateral regions extending hundreds of nanometers from dense projections. Docked vesicles were uniformly depleted after stimulation in wild-type animals (*Figure 3D*). In mutants lacking CaV2 channels, docked vesicles were selectively retained adjacent to the dense projection, but fused normally in regions distal from the dense projection. These results indicate that CaV2 is essential for vesicle fusion at sites adjacent to the dense projection (*Figure 3E*). Complete loss of both CaV2 and CaV1 function in the nervous system is lethal (*Figure 1A*). Therefore, to assay mutation of both channels in the nervous system we used a weak allele of CaV1; the hypomorph *egl-19(n582)* is viable in double mutants with *unc-2(lj1)*. Mutation of the CaV1 channel reduced fusion broadly, although significant vesicle fusions were observed within 100 nm of the dense projection (*Figure 3F*). In the absence of RyR only CaV2 is functional, and vesicle fusions were only observed in the 33 nm pool — directly adjacent to the dense projection (*Figure 3G*). The CaV1 CaV2 double mutant exhibited no change in the number and distribution of docked synaptic vesicles after stimulation (*Figure 3H*). These data indicate that CaV2 and CaV1 act on two spatially distinct pools of synaptic vesicles at the same synapses in *C. elegans*: a central pool dependent on CaV2 calcium channels and a lateral pool dependent on CaV1 and RyR.

To more finely partition the fusion domains for which each channel, we binned micrographs into three zones: adjacent to the dense projection (0–33 nm), intermediate active zone (33–165 nm), and lateral active zone (165–594 nm). Baseline docking was increased in the 0–33 nm bin in the CaV2(-) mutants, this trend is consistent with decreased tonic fusion adjacent to the dense projection (*Figure 3—figure supplement 1A*), and release probability of vesicles is reduced adjacent to the dense projection in CaV2(-) mutants (*Figure 3—figure supplement 1D*).

In the intermediate active zone (33–165 nm), there was no change in the baseline docking of any mutant (*Figure 3—figure supplement 1B*). However, we observed identical fusion defects in the intermediate zone in the CaV2(-) mutant and RyR(-) mutant, which suggests CaV2 activates RyR to release calcium from internal stores. Activation of RyR by CaV2 is consistent with the multiquantal release mediated by these channels (*Figure 2E–G*).

In the distal active zone (165–594 nm), the probability of vesicle fusion was reduced in CaV1(-) mutants, RyR(-) mutants, and CaV2(-) CaV1(-) double mutants, but slightly increased in CaV2(-) mutants (*Figure 3—figure supplement 1D*). An increase in fusion at distal sites in CaV2 mutants might be due to compensatory effects in the expression or organization of CaV1 and RyR.

Together, electron microscopy indicates that CaV2 mediates fusion of vesicles adjacent to the dense projection, and that CaV1 mediates fusion of vesicles at lateral sites. The ryanodine receptor is essential for vesicle fusion mediated by CaV1 at lateral sites and contributes to vesicle fusion by CaV2 near the dense projection.

## CaV2 and CaV1 differentially localized at synapses

The ultrastructural data suggests that distinct calcium channels act at spatially separate areas of the active zone. To determine if these calcium channels are physically located at these sites, we used fluorescence microscopy. We modified the endogenous genes to encode tags for synthetic fluorescent ligands and performed three-color imaging using the dense projection as an anatomical fiducial at the center of the synapse. Because *C. elegans* synaptic varicosities are less than 1 µm in diameter, superresolution microscopy was required to resolve channel clusters. A segment of the dorsal nerve cord was imaged, and the region of imaging was restricted to a narrow band to avoid potential complications by CaV1 expression in muscle. All imaging was conducted on living, acutely anesthetized nematodes.

Multiple tagging sites were tested for all genes, but in some cases the tags disrupted function. Therefore, we tagged internal sites within regions of poor conservation (*Figure 4—figure supplement 1A and B*). For example, CaV2 was tagged with HALO (*Los et al., 2008*) in the second extracellular loop near the N-terminus (*Kurshan et al., 2018*; *Schwartz and Jorgensen, 2016*). The strains used

for three-color imaging exhibited normal morphology and appear to move like wild-type animals, suggesting the tagged proteins are functional. Analysis of specific movements indicated that most locomotory responses are normal; however, the frequency of reversals was increased in all multiply tagged strains (*Figure 4—figure supplement 1E–I*).

To confirm that the pattern of calcium channels in our fluorescence images matched the arrangement of dense projections, we reconstructed 20 μm of the dorsal nerve cord from serial sections with electron microscopy (*Figure 4A*). Dense projections at neuromuscular junctions were spaced roughly 1 μm apart in the reconstruction (1.02 / μm). This matches well to the distribution of CaV2 clusters (1.10+/-0.16 μm) along the dorsal cord by super-resolution microscopy, and is consistent with previous studies demonstrating that CaV2 /UNC-2 channels are localized to dense projections by immuno-electron microscopy (*Gracheva et al., 2008*). To further demonstrate that CaV2 is localized to the dense projection, we compared CaV2 localization with CRISPR-tagged presynaptic active zone proteins, including Neurexin (*nrx-1*), Magi (*magi-1*), SYDE (*syd-1*), Liprin-α (*syd-2*), RIMBP (*rimb-1*), and α-Catulin (*ctn-1*). The endogenous genes were tagged with the fluorescent protein Skylan-S and colocalization assessed with HALO-tagged CaV2 in the dorsal nerve cord of transgenic worms. CaV2 colocalized with all these proteins (*Figure 4B*), indicating that it is indeed localized at the dense projection.

ELKS clusters in particular are tightly associated with CaV2 clusters and exhibit the same 1 μm spacing (*Figure 4C*), and serves as a synaptic fiducial. To quantify the distribution of CaV2 relative to ELKS, the center of mass of the cluster centers was determined, an X-axis was plotted between the two cluster centers, and localizations were placed onto a 2D plot (*Figure 4D*). The ELKS center of mass was defined as the origin, and the distance of each CaV2 localization to the Y-axis of the ELKS cluster was assigned as an axial coordinate. These distances were collapsed onto a 1D plot, distances were binned and frequency plotted (*Figure 4E*). These plots were then used to calculate mean distributions of proteins from multiple synapses. CaV2 clusters and ELKS clusters were similar in diameter (297 nm vs 294 nm, respectively, n=26 synapses). The cluster centers are slightly offset ($\bar{A}$=124 nm), so that 62% of ELKS localizations were within a CaV2 cluster (*Figure 5A–C*). The offset could indicate that the clusters overlap but are not coincident. Alternatively, these proteins may be perfectly colocalized but differ in our plots due to the positions of the tags on the proteins; specifically, CaV2 was tagged on the extracellular side in the synaptic cleft, whereas ELKS was tagged at the C-terminus on the intracellular side. In summary, CaV2 is tightly clustered and associated with an ELKS cluster; the offset may simply be due to tagging sites and the length of the proteins.

In contrast to CaV2, CaV1 was broadly distributed as dispersed puncta in the synaptic bouton ('cluster' diameter 869 nm), and is largely excluded from ELKS and CaV2 clusters (*Figure 5A*). Although dispersed, CaV1 usually exhibits a site of high density that is about 250 nm from the ELKS and CaV2 clusters (262 nm and 274 nm, respectively; *Figure 5C*). Although CaV1 is largely excluded from CaV2 clusters, the clusters often abut one another (*Figure 5B*, see Figure 10 for more examples).

To confirm that the CaV1 localizations are presynaptic and not in the muscle or epidermis, we generated a HALO-tagged CaV1 under the pan-neuronal synaptotagmin promoter (P*snt-1*). This construct was inserted in the CaV1(Δns) strain, and the transgene fully rescues locomotion and behavior (*Figure 1B–I*). For convenience of genetic crosses, we used RIM binding-protein (RIMBP/RIMB-1) as the dense projection marker. The overexpressed CaV1::HALO tended to be more punctate than the endogenously tagged protein (*Figure 5—figure supplement 1A and B*). CaV1 did not colocalize with RIMBP and the mean distance to the dense projection was similar to the endogenously tagged gene (endogenous CaV1 to ELKS, 262 nm; transgene CaV1 to RIMBP, 378 nm; *Figure 5—figure supplement 1E*). To demonstrate that CaV1 clusters are presynaptic as opposed to postsynaptic, we searched for potential binding partners that might colocalize with the CaV1 puncta. SHN-1(Shank) binds the C-terminus of CaV1 via its PDZ domain but is primarily postsynaptic (*Pym et al., 2017*). LIN-7 also contains a PDZ domain but is presynaptic and could serve as a scaffold for CaV1 (*Butz et al., 1998*; *Hallam et al., 2002*). We expressed LIN-7 in acetylcholine motor neurons using the *unc-129* promoter (*Figure 5—figure supplement 1D*). CaV1 and LIN-7 localizations were overlapping, but distal to the dense projection marker RIMBP (*Figure 5—figure supplement 1E*). These data suggest that CaV1 is localized at presynaptic boutons in a separate domain from CaV2 channels.

If CaV1 and RyR function in the same vesicle fusion pathway as the physiology data suggests, they should be colocalized in motor neurons (*Piggott and Jin, 2021*). RyR was tagged with HALO at the N-terminus of the neuronal isoform (*Figure 4—figure supplement 1C*; *Marques et al., 2020*). RyR

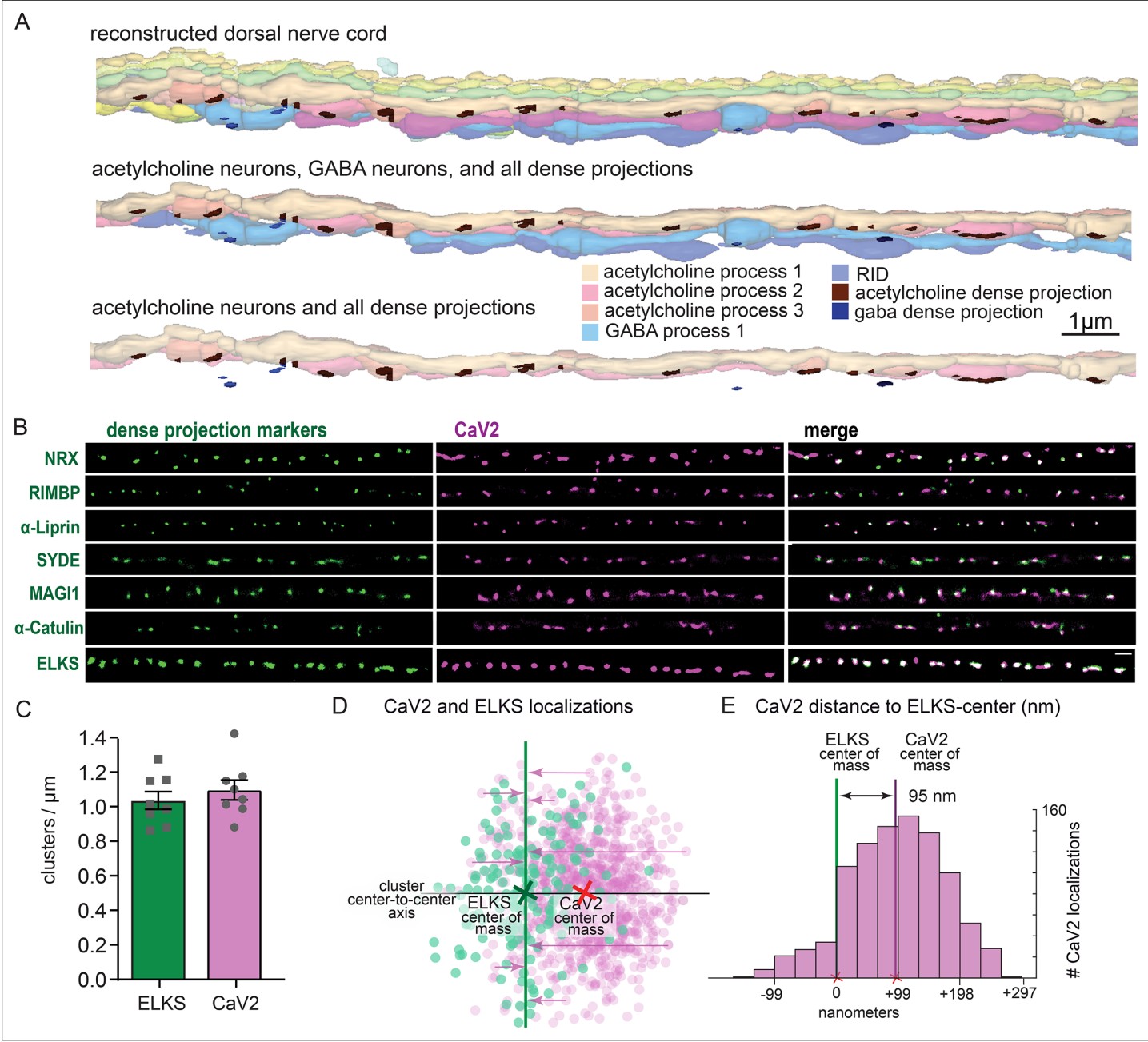

**Figure 4.** Dorsal nerve cord reconstruction and candidate dense projection markers. (**A**) 20 micron reconstruction of the wild-type *C. elegans* dorsal nerve cord. Dense projections are highlighted to compare to superresolution images below. Scale bar 1 μm, section thickness 100 nm. (**B**) CaV2 colocalizes with cytomatrix active zone proteins. Super-resolution images of Skylan-S-tagged cytomatrix protein homologs in *C. elegans* NRX-1, RIMB-1, SYD-2, SYD-1, MAGI-1, CTN-1, ELKS-1 compared to CaV2-HALO in the same animal. (**C**) ELKS and CaV2 clusters form approximately 1 / μm along the dorsal nerve cord from super-resolution image analysis. Clusters were quantified for over dorsal nerve cords with an average length of 17.8 μm, N=8 animals (**D**) Localization plot tool (Proberuler) example diagram of a single ELKS (green) and CaV2 (purple) synapse. Cluster centers are marked by solid lines. (**E**) Histogram of CaV2 localization distance to ELKS center of mass axis from example ELKS and CaV2 synapse.

The online version of this article includes the following figure supplement(s) for figure 4:

**Figure supplement 1.** Tagging sites for calcium channels and UNC-13.

localizations were compared to CaV1 localizations and the dense projection marker ELKS (*Figure 6A and B*). RyR localizations were diffusely distributed, and lateral to the dense projection (ELKS to RyR center of mass distance: 393 nm; 25 synapses) (*Figure 6C*). RyR localizations were correlated with CaV1 (RyR to CaV1 center of mass: 166 nm). Visual inspection of the images suggested that RyR and

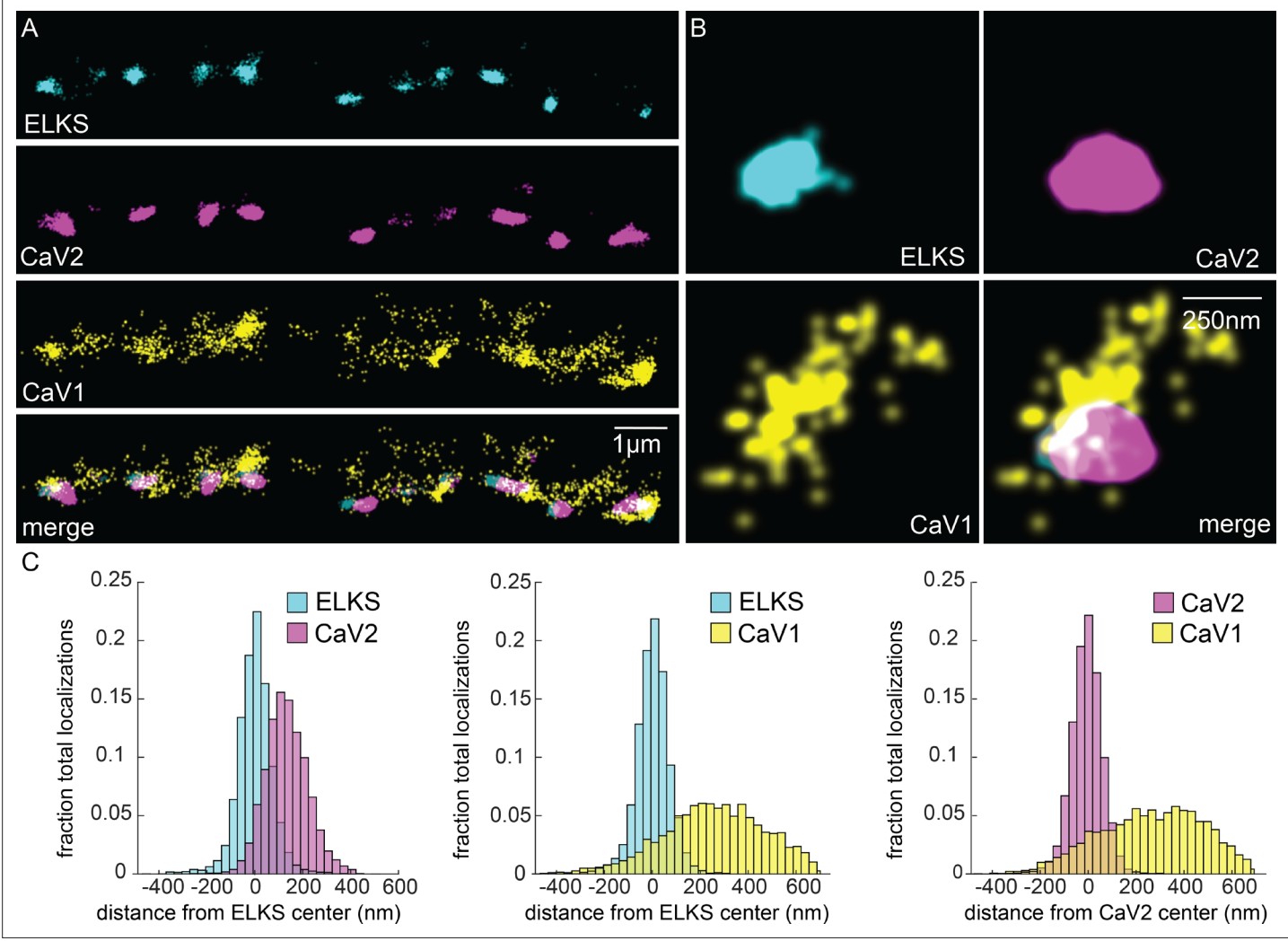

**Figure 5.** CaV1 is excluded from the dense projection, and dispersed in the active zone. (**A, B**) Localization microscopy plots of the dorsal nerve cord. ELKS is tagged with Skylan-S. The CaV2-HALO ligand is HTL-JF646, and the CaV1-SNAP tag ligand is STL-JF549pa. (**A**) CaV2 (magenta) colocalizes with dense projections labeled with ELKS (cyan). CaV1-SNAP (yellow) is largely excluded from the dense projection; and scattered in the synaptic varicosity. Scale bar = 1 µm. (**B**) Distributions of CaV2 and CaV1 in a synapse. Dense projections labeled with ELKS (cyan) colocalize with CaV2 (magenta), but not CaV1 (yellow). Scale bar = 250 nm. (**C**) Quantitation of protein localizations from multiple synapses. The center of mass of localizations was calculated from 2D plots. An axis between the centers was fixed and all localizations collapsed onto the axis. Localizations were combined into 33 nm bins, to match the electron microscopy analysis, and plotted as the fraction of total localizations. Data were collected and combined from n=26 synapses, N=5 animals. Data available as *Figure 5—source data 1*.

The online version of this article includes the following source data and figure supplement(s) for figure 5:

**Source data 1.** Particle files for ELKS/CaV1/CaV2 SMLM data and distance measurements.

**Figure supplement 1.** Tagged CaV1 expressed in neurons forms clusters at presynaptic boutons.

**Figure supplement 1—source data 1.** Particle files for RIMBP/CaV1/LIN-7 SMLM data and distance measurements.

CaV1 are often interdigitated in adjacent zones (*Figure 6A*). To characterize this relationship a nearest neighbor analysis was performed and revealed that 94% of RyR localizations were within 100 nm of a CaV1 localization (*Figure 6D*). CaV1 exhibits a slightly broader distribution; nevertheless, 82% of CaV1 localizations were within 100 nm of a RyR channel. The spatial correlation between CaV1 and RyR is consistent with the functional coupling observed by physiology and electron microscopy.

## Different UNC-13 isoforms are associated with CaV1 and CaV2

Vesicle docking and SNARE priming require UNC-13 proteins. Null mutations in *unc-13* nearly eliminate neurotransmission and vesicle docking in *C. elegans* (*Hammarlund et al., 2007*; *Richmond*

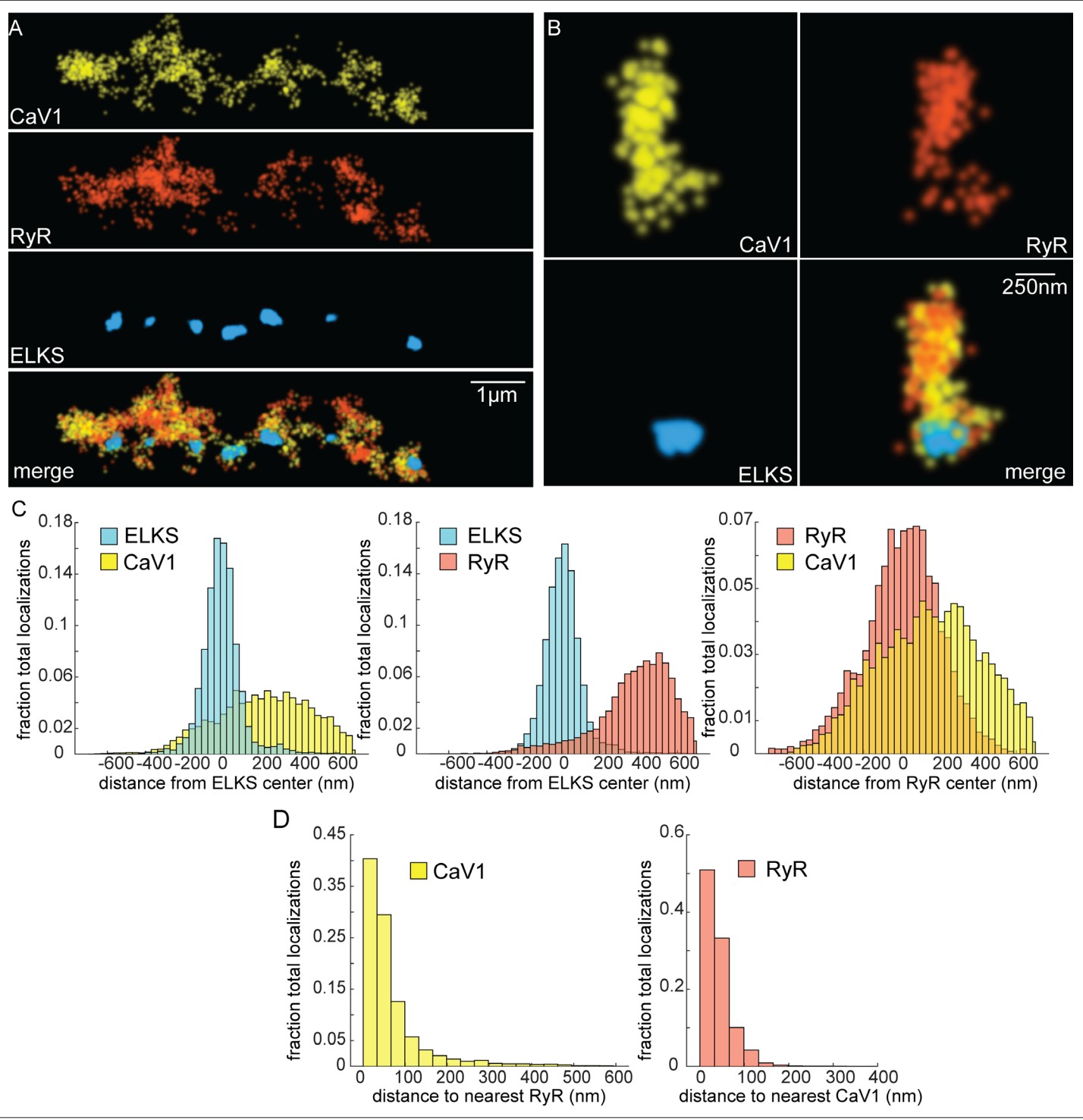

**Figure 6.** CaV1 and RyR are adjacent. (**A**) CaV1 and RyR are adjacent along the dorsal nerve cord, lateral to the dense projection. Animals and HTL-JF646. Scale bar = 1 μm. CaV1-SNAP is labelled with STL-JF549pa, RyR-HALO is labelled with HTL-JF646, and dense projections are labeled by ELKS-Skylan-S. (**B**) RyR and CaV1 colocalize within synapses. Labelling as in 'A'. Scale bar = 250 nm. (**C**) Distances from CaV1-SNAP localizations to center of ELKS-Skylan-S cluster versus ELKS localizations to ELKS center. Distances from RyR-HALO localizations to center of ELKS-Skylan-S cluster versus ELKS localizations to ELKS center. Distances from CaV1-SNAP localizations to the center of the RyR-HALO cluster versus RyR-HALO localizations to the RyR center. N=5 animals, n=25 synapses. (**D**) RyR and CaV1 are adjacent. Left, nearest neighbor analysis was performed on CaV1-SNAP localizations to find the nearest RyR-HALO localization. Right, nearest neighbor distances from RyR-HALO to CaV1-SNAP were calculated. n=5 animals, 25 synapses. Data available as *Figure 6—source data 1*.

*Figure 6 continued on next page*

*Figure 6 continued*

The online version of this article includes the following source data for figure 6:

**Source data 1.** Particle files for ELKS/CaV1/RyR SMLM data and distance measurements.

*et al., 1999*). In most organisms, there are two types of Unc13 proteins, those with an N-terminal C2A domain, and those lacking the C2A domain (*Dittman, 2019*). Binding of the C2A domain to the scaffolding protein RIM activates Unc13 (*Betz et al., 2001*; *Hu et al., 2013*; *Liu et al., 2019*; *Lu et al., 2006*; *Zhou et al., 2013*). Unc13 isoforms which lack a C2A domain bind ELKS / CAST in flies and mice (*Böhme et al., 2018*; *Kawabe et al., 2017*). In *C. elegans*, the *unc-13* gene encodes two major splice isoforms: a long isoform UNC-13L with a C2A domain, and a short isoform UNC-13S lacking a C2A domain. To determine whether UNC-13 colocalizes with CaV1 and CaV2, we edited the *unc-13* locus to append Skylan-S to the C-termini of both major isoforms ('UNC-13all') (*Figure 4— figure supplement 1D*). Both CaV2 and CaV1 calcium channels were tightly associated with UNC-13 isoforms (*Figure 7A–C*). Nearest-neighbor analysis indicates that 99.7% of CaV2 channels were within 100 nm of an UNC-13 localization, and 89% of CaV1 channels were within 100 nm of an UNC-13 protein (*Figure 7D*).

To determine if UNC-13 isoforms are differentially associated with calcium channels, we tagged the N-terminus of UNC-13S with Skylan-S. UNC-13S did not colocalize with CaV2 (peak-to-peak 319 nm) but was associated with CaV1 (*Figure 8A–C*). Nearest neighbor analysis indicates that 99% of UNC-13S localizations are within 100 nm of a CaV1 channel (*Figure 8D*). These data demonstrate that CaV1 channels are associated with UNC-13S at lateral sites. Although it is possible that UNC-13L is also at these lateral sites, UNC-13S can dock vesicles independent of UNC-13L. Rescue of *unc-13* null animals with UNC-13S restores locomotion and mini frequency to about half of wild-type (*Hu et al., 2013*). In contrast to the null, mutants lacking UNC-13L have normal or elevated numbers of docked vesicles in lateral regions of the synapse (*Hammarlund et al., 2007*; *Zhou et al., 2013*), indicating that the UNC-13S isoform is capable of docking synaptic vesicles in the absence of UNC-13L (*Hu et al., 2013*). Together, the localization data, electron microscopy, and physiology indicate that UNC-13L is coupled to CaV2 at the dense projection, and UNC-13S is coupled to CaV1 and the ryanodine receptor at lateral sites.

## Discussion

Calcium channel classes tend to be associated with specific tissue functions: CaV2 (N, P/Q, R-type) with synaptic transmission, and CaV1 (L-type) channels with muscle contraction. Here, we demonstrate that both CaV2 and CaV1 channels drive vesicle fusion at *C. elegans* neuromuscular junctions and mediate the release of different synaptic vesicle pools. In electrophysiological assays, these pools are genetically separable and complementary. Flash-and-freeze electron microscopy revealed that CaV2 channels fuse vesicles near the dense projection at the center of the synapse; whereas CaV1 channels fuse vesicles at lateral sites in the same synapses. Super-resolution imaging indicates that CaV2 channels are compacted at the dense projection into 250 nm clusters, along with the active zone proteins ELKS, neurexin, α-Liprin, and RIMBP. CaV2 is associated with the long isoform of the docking and priming protein UNC-13L. By contrast, CaV1 is dispersed in the synaptic varicosity and is associated with the short isoform UNC-13S. Finally, vesicle fusion mediated by CaV1 is dependent on the ryanodine receptor, presumably to activate calcium release from internal stores (*Figure 9*).

Participation of multiple classes of calcium channels at the same synapse may serve to tune the dynamics of neurotransmission (*Dolphin, 2021*). Different calcium sources could regulate the strength of output, dynamics of release, or even termination of synaptic activity. Below we discuss the potential contributions of channel numbers, clustering, distance to docked vesicles, and voltage-dependence to differential synaptic behavior.

### Counting calcium channels

The number of calcium channels per synapse can be estimated from the single molecule localization data (see Methods for details). CaV1 channels are dispersed in the synapse but often also form small clusters (*Figure 10A, F and G*). The number of CaV1 channels was calculated from the mean number

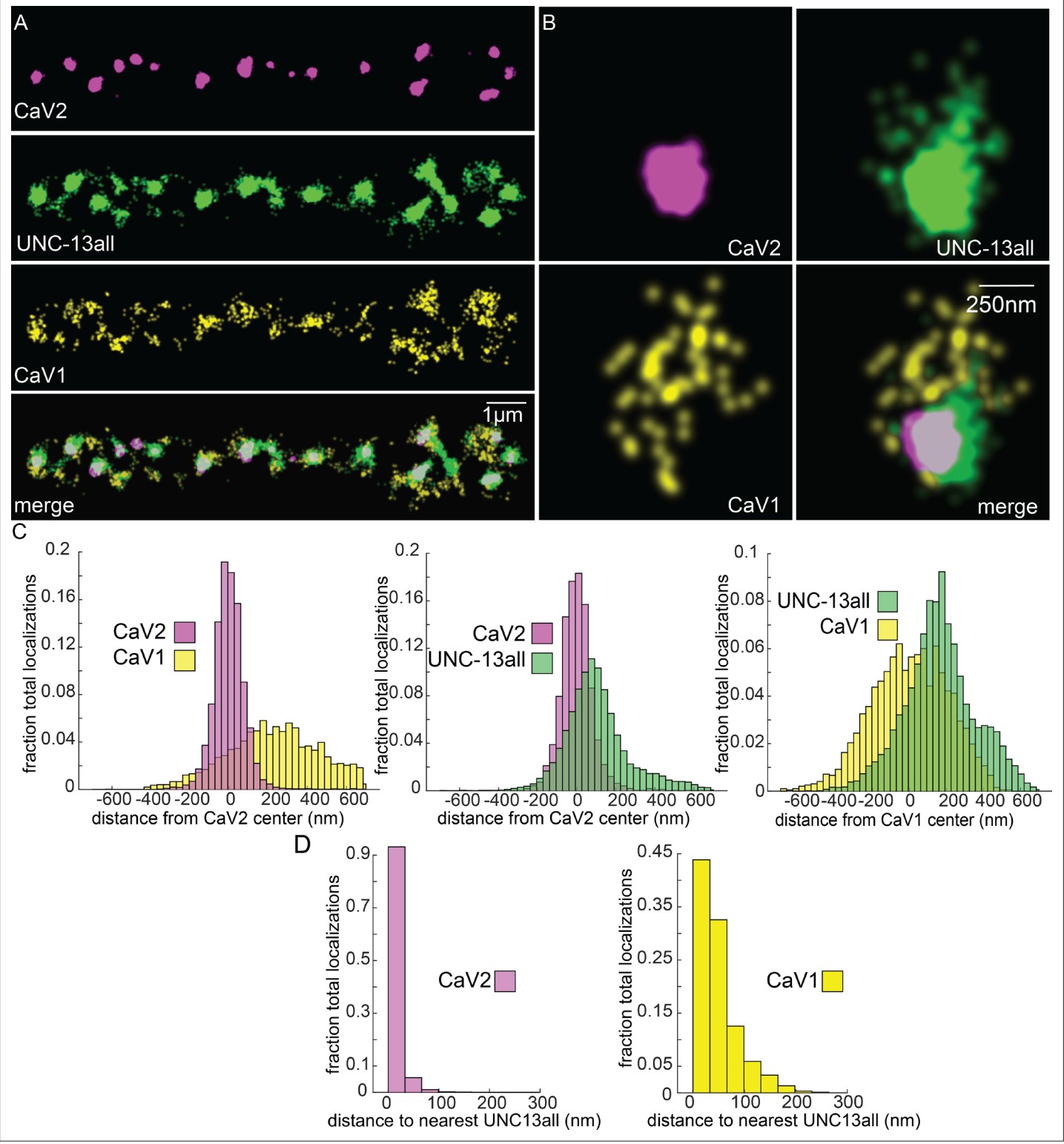

**Figure 7.** UNC-13 isoforms colocalize with CaV2 and CaV1 calcium channels. Localization microscopy identifies CaV1 and CaV2 associated with 'UNC13all', which labels a C-terminal site common to all UNC-13 isoforms. (**A**) UNC-13all colocalizes with CaV1 and CaV2 along the dorsal nerve cord. Proteins are labelled with CaV2-HALO stained with HTL-JF646, CaV1-SNAP stained with STL-JF549, and UNC13all-Skylan-S. (**B**) UNC-13all colocalizes with CaV1 and CaV2 within synapses. Staining as in 'A'. (**C**) Left, distances from CaV1-SNAP localizations to the center of the CaV2-HALO cluster, and CaV2-HALO localizations to the center of the CaV2-HALO cluster. Middle, distances from UNC13all-Skylan-S localizations to the center of the CaV2-HALO cluster. Right, distances from UNC13all-Skylan-S localizations to the center of the CaV1-SNAP cluster, n=5 animals, 25 synapses. (**D**) Left, nearest-

*Figure 7 continued on next page*

*Figure 7 continued*

neighbor distances between UNC13all and CaV1 and CaV2 localizations. Right, nearest neighbor analysis between UNC13all-Skylan-S and CaV2-HALO or CaV1-SNAP measured from synaptic regions, n=5 animals, 25 synapses. Data available as *Figure 7—source data 1*.

The online version of this article includes the following source data for figure 7:

**Source data 1.** Particle files for UNC-13all/CaV1/CaV2 SMLM data and distance measurements.

of blinks per channel, as well as from the total photon flux, converging on 77±15 CaV1 channels per synapse (*Figure 10B–E*).

The ryanodine receptor images were suffused with high background fluorescence, and photon flux was not a reliable measure. Using the mean blinks per channel produces an estimate of 29±4 RYR channels per synapse (*Figure 10G*).

CaV2 channels are tightly localized to the dense projection (*Figure 10F*). In our images, the cluster appears as a solid mass; the overlap in localization precision made it impossible to assign blinks or photon flux to individual channels in the cluster. However, assuming the blinking rate of rhodamine dyes is similar to cyanine dyes (*Helmerich et al., 2022*), the frequency of blinking indicates that the cluster contains 101±16 CaV2 channels.

The ~100 CaV2 channels per synapse derived from our single molecule localization data is much higher than the ~35 CaV2.1 channels determined by immunogold labelling synapses in the mouse central nervous system (*Holderith et al., 2012*; *Kusch et al., 2018*). Nevertheless, the density of calcium channels at *C. elegans* neuromuscular junctions (91 CaV2 per $\mu m^2$) is similar to mammalian synapses (100–400 CaV2.1 channels per $\mu m^2$).

## High-density CaV2 clusters mediate rapid fusion

The dense cluster of CaV2 channels likely insures reliable synchronous fusion, due to the large number of channels and because of the tight coupling distance to docked vesicles. A large number of channels is required because single calcium channels open stochastically, and can introduce jitter to the precise timing of a signal (*Borst and Sakmann, 1996*). For synapses to reliably track high frequency action potentials, there must be a sufficient number of channels to insure that some open immediately in response to depolarization. Nematode motor neurons rely on graded potentials rather than conventional action potentials; nevertheless, a dense cluster of CaV2 channels will promote a rapid synaptic response to rapid depolarizations.

The calcium channels must be physically coupled to the neurotransmitter release site, that is, the docked synaptic vesicle must be just 20–30 nm from the calcium channel, for the calcium concentration in the nanodomain to be high enough to drive vesicle fusion (*Fedchyshyn and Wang, 2005*; *Weber et al., 2010*). Synaptic vesicle pools can be designated as 'tightly coupled' or 'loosely coupled' to calcium channels based on their sensitivity to EGTA (*Dittman and Ryan, 2019*; *Eggermann et al., 2011*). At *C. elegans* neuromuscular junctions, UNC-13L mediates tight coupling (EGTA-insensitive), whereas UNC-13S mediates loose coupling (EGTA-sensitive) (*Hu et al., 2013*). Here, we found that UNC-13L is colocalized with CaV2 at dense projections. Consistent with EGTA-insensitive priming by UNC-13L, CaV2 mediates the release of vesicles within 33 nm of the dense projection (*Hammarlund et al., 2007*).

Finally, the dense cluster of CaV2 channels transfers enough calcium from the synaptic cleft to fuse multiple vesicles upon depolarization – termed multiquantal release. The calcium influx from CaV2 also activates the ryanodine receptor, which in turn drives fusion of vesicles docked beyond 33 nm in the intermediate active zone (33–165 nm). The frequency of CaV2-mediated fusion is not affected by presence or loss of RyR, indicating that CaV2 reliably drives vesicle fusion in response to depolarizations on its own. Only the amplitude of these currents is reduced in the absence of RyR, indicating that the ryanodine receptor only potentiates CaV2 responses by releasing calcium from internal stores.

## CaV1 requires coupling to the ryanodine receptor

In contrast to CaV2, CaV1 localizations are dispersed broadly in the synapse and localizations are frequently solitary. Calcium influx mediated by CaV1 is not sufficient to drive vesicle fusion directly, but the ryanodine receptor is activated by low levels of cytosolic calcium and releases calcium from internal stores. CaV1 and RyR colocalize with the UNC-13S synaptic vesicle docking protein. Two-step

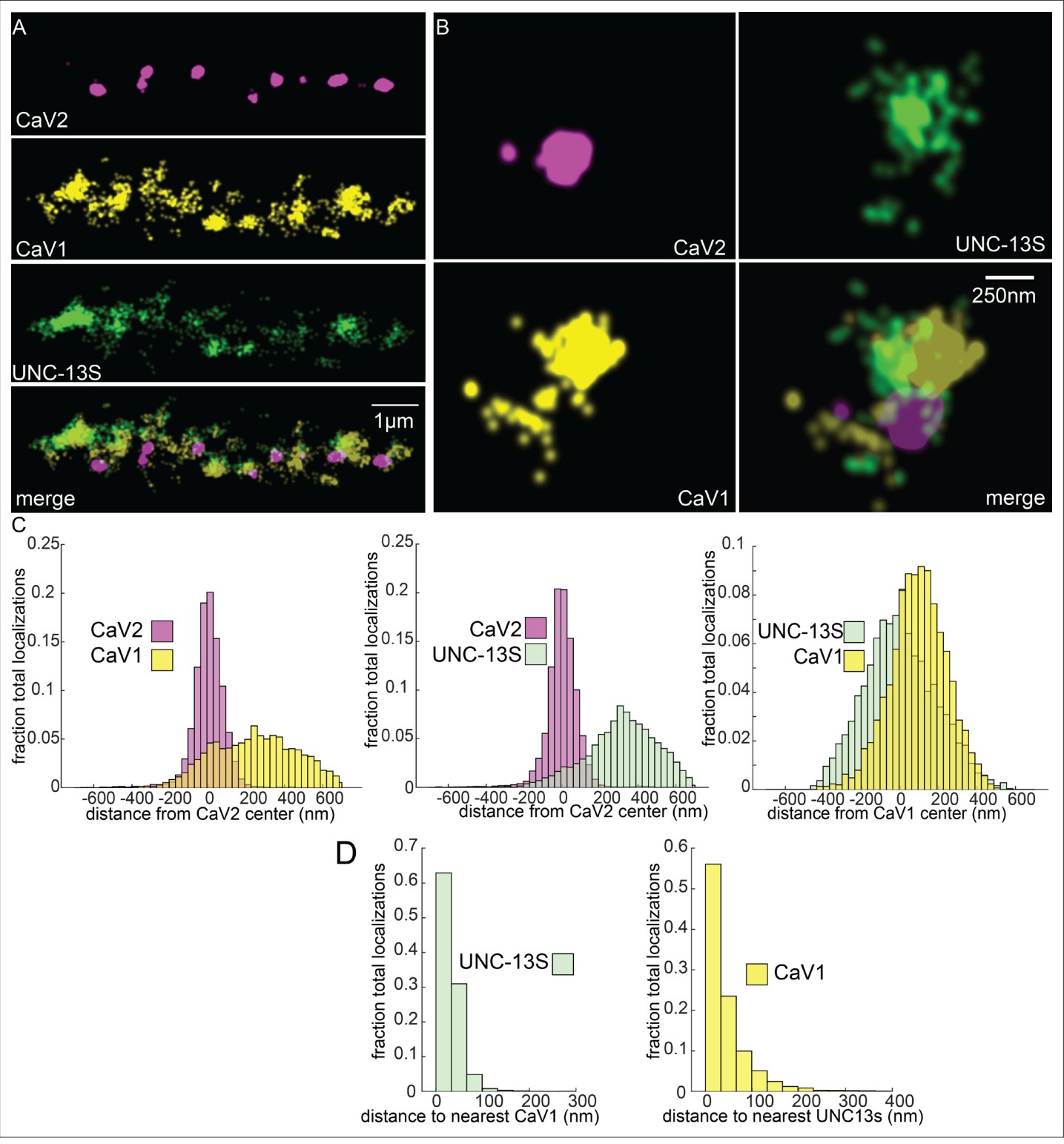

**Figure 8.** UNC-13S is associated with CaV1 calcium channels. Localization microscopy identifies CaV1 associated with UNC-13S which labels a n-terminal site common to a short isoform. (**A**) UNC-13S localizes with CaV1 along the dorsal cord, but not with CaV2. The endogenous protein tags CaV2-HALO was stained with HTL-JF646, CaV1-SNAP with STL-JF549, and imaged with Skylan-S-UNC-13S using single-molecule localization microscopy. (**B**) UNC-13S localizes with CaV1 within synapses. Strain was labelled as in 'A'. (**C**) Left, distances from CaV1-SNAP localizations to the center of the CaV2-HALO cluster compared to CaV2-HALO localizations to their own center. Middle, distances from Skylan-S-UNC-13S localizations to the center of the CaV2-HALO cluster. Right, distances from Skylan-S-UNC-13S localizations to the center of the CaV1-SNAP cluster. N=5 animals, 25 synapses

*Figure 8 continued on next page*

*Figure 8 continued*

(**D**) Nearest-neighbor distances of CaV1-SNAP to Skylan-S-UNC-13S localizations. Nearest neighbor analysis of Skylan-S-UNC-13S to CaV1-SNAP measured from synaptic regions. N=5 animals, 25 synapses. Data available as *Figure 8—source data 1*.

The online version of this article includes the following source data for figure 8:

**Source data 1.** Particle files for UNC-13S/CaV1/CaV2 SMLM data and distance measurements.

calcium signaling is consistent with the EGTA-sensitivity of UNC-13S vesicle fusion (*Hu et al., 2013*); EGTA can buffer the small amount of calcium from CaV1 before it detonates the ryanodine receptor, or buffer calcium diffusing from the endoplasmic reticulum to the UNC-13S docking site.

CaV1 channels inactivate slowly (the moniker 'L-type' refers to 'long-lasting') and are therefore more responsive to long duration synaptic depolarizations (*Naranjo et al., 2015*; *Yu et al., 2018*). For example, slow inactivation of CaV1.3 and CaV1.4 channels allows synapses in sensory neurons to accurately report the depolarization status of the synaptic bouton (*McRory et al., 2004*; *Platzer et al., 2000*). Similarly, EGL-19 exhibits slow inactivation (*Lainé et al., 2014*), which likely contributes to graded miniature currents at *C. elegans* synapses (*Liu et al., 2018b*).

## The ryanodine receptor mediates multiquantal release

CaV2 and CaV1 both contribute to activation of RyR, but all transmission at lateral sites by CaV1 relies on activation of RyR. The miniature currents mediated by RyR are large amplitude currents, and are likely to be multiquantal vesicle fusions. However, we cannot exclude the possibility that the receptor

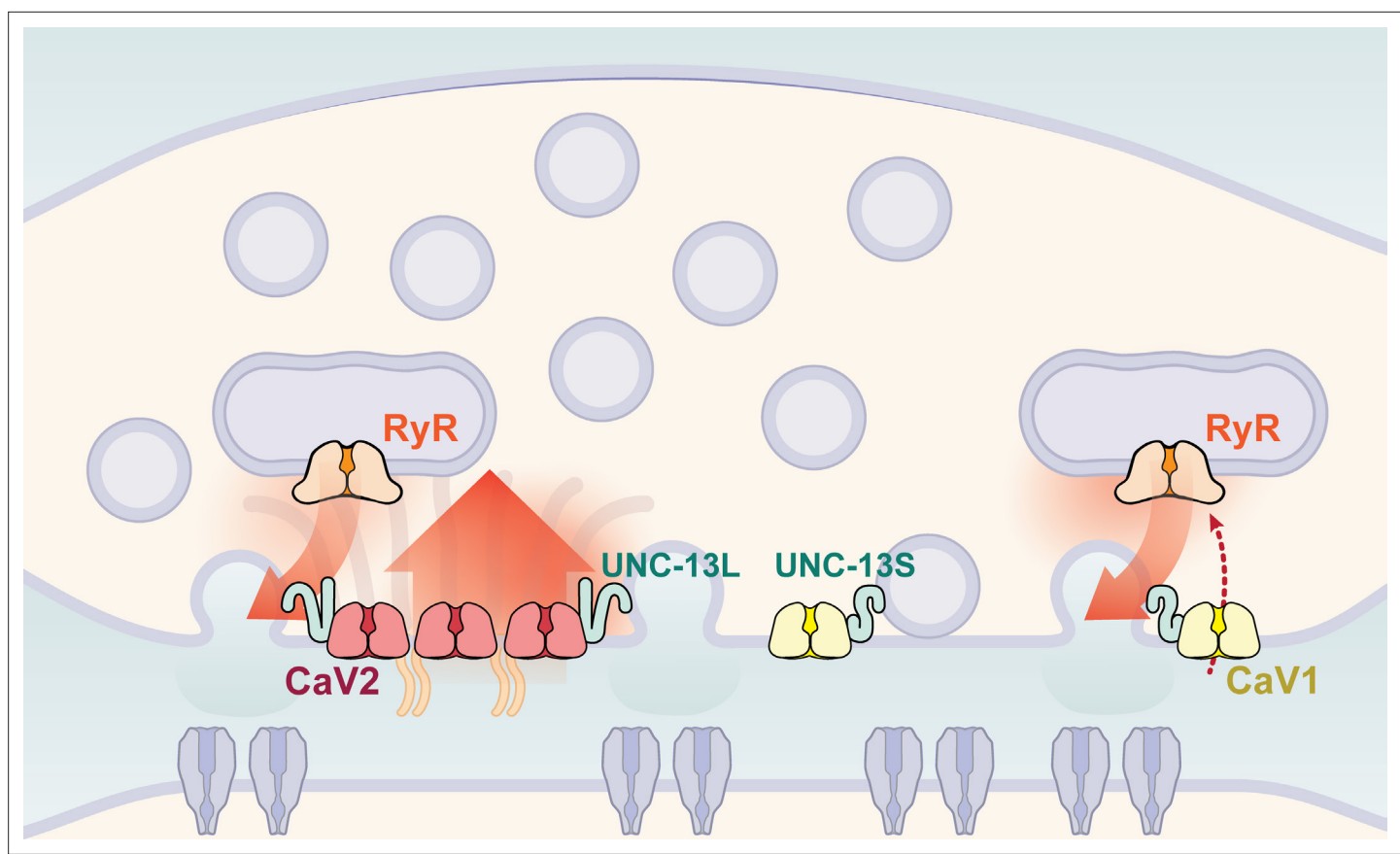

**Figure 9.** Two independent release sites for synaptic vesicles. Voltage-gated calcium channels localize to two distinct zones at the neuromuscular synapse of *C. elegans*. The CaV2 channel localizes to the dense projection along with ELKS, RIMBP, Neurexin, Liprin-alpha, SYDE, MAGI1, alpha-Catulin and the SNARE priming protein UNC-13L. CaV2 is required to fuse synaptic vesicles are docked directly adjacent to the dense projection. The second channel CaV1 is at a lateral site centered ~300 nm from the dense projection but can span hundreds of nanometers. CaV1 requires coupling to RyR to synaptic vesicles at the lateral site. These near and far pools utilize specific release machinery. Most UNC-13all localizes to the dense projection. However, some UNC-13 localizes with CaV1 at the lateral site. Isoform specific tagging shows UNC-13S localized with lateral site.

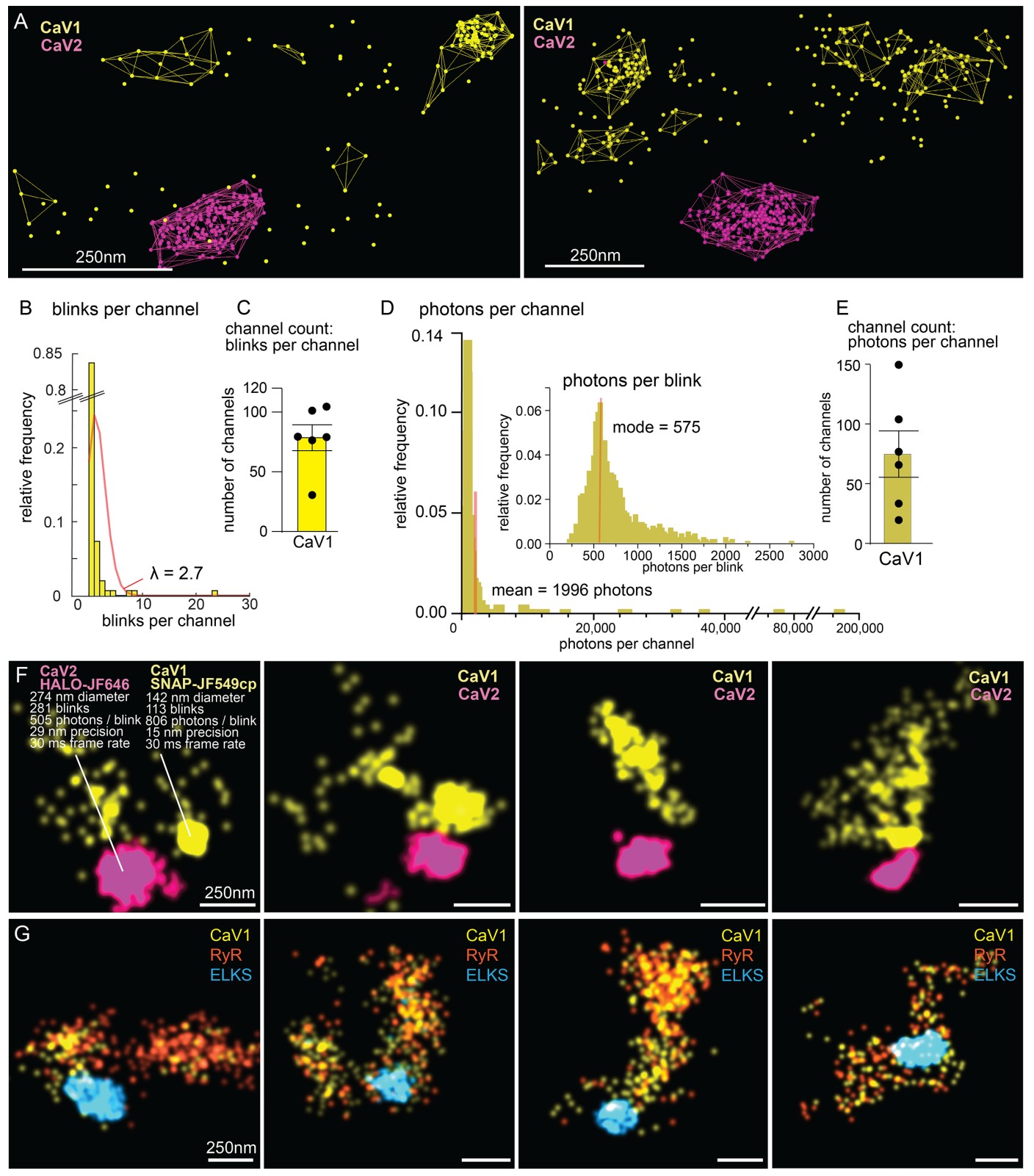

**Figure 10.** Counting calcium channels and example synapses. (**A**) Superresolution images of *C. elegans* expressing CaV1::SNAP, CaV2::HALO. Animals were stained with HTL-JF646, STL-JF549cp. Clusters of calcium channels are indicated with colored wireframes. Scale bar = 250 nm. (**B**) Poisson distribution fitted to frequency distribution plot of number of blinks per channel. (**C**) CaV1 channel count based on number of total number of blink over blinks per channel. (**D**) Frequency distribution of number of photons per channel cluster. Inlay: Frequency distribution of number of photons per

*Figure 10 continued on next page*

*Figure 10 continued*

blink. (**E**) CaV1 channel count based on number of total photons over the mode of photons per channel. (**F**) Four example superresolution images of *C. elegans* expressing CaV1::SNAP, CaV2::HALO. Animals were stained with HTL-JF646, STL-JF549cp. A sample cluster of CaV1 was 142 nm in diameter, contained 113 blinks that on average emitted 806 photons. These blinks were localized with 15 nm precision at a 30ms frame rate. A sample CaV2 cluster was 274 nm in diameter, contained 281 blinks with an average emissions of 505 photons per blink. Blinks were localized with 29 nm precision at a 30ms frame rate. (**G**) Four example superresolution images of *C. elegans* expressing CaV1::SNAP, RyR::HALO. Animals were stained with HTL-JF646, STL-JF549pa. n=6 synapses N=3 animals. Error bars reported in SEM. CaV1 data available as *Figure 10—source data 1*. CaV2 data available as *Figure 10—source data 2*.

The online version of this article includes the following source data for figure 10:

**Source data 1.** Quantification of CaV1 channels by spatial constraint and photon flux.

**Source data 2.** Particle files of CaV2 localization data for channel count estimations.

field under the lateral site is larger than the field at the dense projection, or that lateral synaptic vesicles contain more neurotransmitter.

Despite the potentiation of neurotransmitter release by the ryanodine receptor, the ultimate output of calcium release from internal stores might be a rapid shutdown of synaptic transmission. In CA1 hippocampal neurons CaV1 channels and RyR2 activate calcium gated-potassium channels in the cell body of mouse CA1 hippocampal neurons and hyperpolarize the neuron (*Sahu et al., 2019*). In *C. elegans*, CaV1 activity is specifically coupled to SLO-2 BK potassium channels (*Liu et al., 2014*), activation of SLO-2 by a calcium burst would hyperpolarize the membrane and terminate synaptic transmission. Furthermore, RyR is inhibited by high concentrations of calcium. Hyperpolarization mediated by calcium sparks could act as a safety mechanism to break the positive feedback loop of calcium at synapses, or could regulate switching between neuronal circuits.

The ryanodine receptor is directly stimulated by calcium binding to the receptor (*des Georges et al., 2016*). Activation of the ryanodine receptor will then depend on the diffusion rate of calcium, buffering capacity, and the distance between the voltage-gated calcium channel and the ryanodine receptor. Modeling studies suggest that the endoplasmic reticulum must be within ~100 nm of vesicle, and that high-frequency stimulation may be required to overcome these barriers (*Bouchard et al., 2003*).

Alternatively, the voltage sensor in CaV1 could be physically coupled to RyR and bypass the requirement for calcium diffusion. In skeletal muscle, CaV1.1 is physically coupled to RYR1 and voltage-sensing by the calcium channel can gate the ryanodine receptor in the absence of extracellular calcium (*Schneider, 1994*; *Shoshan-Barmatz and Ashley, 1998*). In hippocampal cells, CaV1.3 is physically coupled to RyR2 and depolarization alone is sufficient to release calcium stores (*Kim et al., 2007*). Similarly, in *C. elegans*, large amplitude currents were not eliminated in 0 mM extracellular calcium (*Liu et al., 2005*). But in *unc-68(-)* mutants, which lack the ryanodine receptor, miniature currents were completely eliminated in 0 mM external calcium (*Liu et al., 2005*). These data suggest that in worms, release of calcium from internal stores may also be coupled by a physical link to a voltage sensor.

## Active zones at invertebrate and vertebrate synapses

Invertebrate synapses are marked by a prominent dense projection ('T-bar' in flies) that is not observed at vertebrate synapses, with the exception of inner ear hair cell. Our data indicate that this structure comprises many of the proteins associated with the full breadth of the vertebrate synapse, including the adhesion molecules Neurexin and SYDE, and the active zone proteins RIMB and ELKS / CAST. Although lacking a focused density, the vertebrate synapse is not homogenous in organization; CaV2, RIM, Munc13, and Bassoon form irregular clusters in the active zone (*Holderith et al., 2012*; *Kawabe et al., 2017*; *Kusch et al., 2018*; *Tang et al., 2016*). The invertebrate dense projection may serve to create a single density of CaV2 channels for synaptic reliability and speed at the sacrifice of synaptic plasticity.

Participation of the CaV1-RyR axis at synapses may be widespread. In addition to CaV2 channels, CaV1 channels are also present at neuromuscular junctions in the fly and mouse (*Katz et al., 1996*; *Krick et al., 2021*; *Urbano and Uchitel, 1999*). CaV1.2 and CaV1.3 are also expressed broadly in the brain (*Hell et al., 1996*; *Hell et al., 1993*; *Nanou and Catterall, 2018*). Pharmacological experiments suggest that CaV2 and CaV1 channels function together in GABA neurons in the central nervous system. Moreover, these synapses depend on both 'tight' and 'loose' coupling to calcium channels

as assayed by EGTA (*Eggermann et al., 2011*; *Goswami et al., 2012*; *Rey et al., 2020*; *Vyleta and Jonas, 2014*). CaV1 could also be linked to the release of internal stores in the central nervous system, since ryanodine receptors are found at vertebrate presynapses (*Bouchard et al., 2003*). These ryanodine receptors play a role in multiquantal release at inhibitory inputs to Purkinje cells, and in hippocampal slice culture (*Llano et al., 2000*; *Sharma and Vijayaraghavan, 2003*). Recruitment of the CaV1-RyR axis would increase neurotransmitter release from these synapses, but ultimately might shut down neurotransmission by activating potassium channels. The tuning of the output of the presynapse then ultimately depends on the timing and spatial organization of calcium at release sites (*Eggermann et al., 2011*; *Nakamura et al., 2015*).

In *C. elegans*, these calcium channels regulate two separate synaptic vesicle pools. It is not known whether the composition of the vesicles in these pools differ, as has been observed in vertebrates (*Kavalali, 2015*). In *C. elegans*, these pools are distinguished by the docking and priming machinery: Vesicle docking at CaV2 sites is mediated by UNC-13L; docking to CaV1 sites is mediated by UNC-13S. This organization could be conserved at vertebrate synapses; a subset of hippocampal synapses express orthologs of both UNC-13L (Munc13-1) and UNC-13S (bMunc13-2) (*Kawabe et al., 2017*). Further studies are required to test whether CaV2 is coupled to Munc13-1, and the CaV1-RyR axis is coupled to bMunc13-2 in vertebrates, and eventually to determine how these different release sites regulate synapse kinetics and circuit behavior.

## Methods
### Rescue of lethal calcium channel mutants

Lethal CaV1 /*egl-19(st556)* animals were rescued by Mos-mediated transgenes (*oxTi1047[Pset-18::egl-19b::let-858 3'utr] II.* EG9034 'CaV1 (Δns)') or by extrachromosomal array (*oxEx2017[Pset-18::eGFP_egl-19b::let858utr; Punc-122::GFP]*. EG8827 'CaV1(Δns) RyR(-)') (*Frøkjær-Jensen et al., 2014*). An *egl-19* minigene was constructed from cDNA and portions of gDNA containing small introns to aid expression. The first exons 1–4 are cDNA, followed by gDNA of exon 5–9, and cDNA of exon 10–17. The minigene was placed downstream from a muscle P*set-18* promoter and inserted directly into the genome by MosSCI (*Frøkjaer-Jensen et al., 2008*).

For the array rescue of CaV1 in muscle, *Pset-18::eGFP_egl-19b::let858utr; Punc-122::GFP* was microinjected into the gonad of adult hermaphrodite *egl-19(n582) C. elegans*. Array positive animals were selected and crossed with *egl-19(st556)* (RW3563), which rescued lethality but lacked expression

**Table 1.** Summary of strain nomenclature and alleles.

| Common name | Usage | CaV2 | CaV1 | RyR | CaV2 rescue | CaV1 rescue |
|---|---|---|---|---|---|---|
| Wildtype N2 Bristol | ePhys EM behavior | wt | wt | wt | n/a | n/a |
| CaV1(rescue) | behavior SMLM | wt | st556 | wt | n/a | oxTi1047[Pset-18::egl-19b] oxTi1049[Psnt-1::HALO::egl-19b] |
| CaV1(Δns) | behavior ePhys | wt | st556 | wt | n/a | oxTi1047[Pset-18::egl-19b] |
| CaV1(Δns) RyR(-) | ePhys | wt | st556 | e540 | n/a | oxEx2017[Pset-18::eGFP::egl-19b] |
| CaV2(Δnmj) CaV1(Δns) | ePhys | lj1 | st556 | wt | oxEx2096[Punc-17h::SNAP::unc-2] | oxTi1047[Pset-18::egl-19b] |
| CaV2(Δnmj) RyR(-) | ePhys | lj1 | wt | e540 | oxEx2096[Punc-17h::SNAP::unc-2] | n/a |
| CaV2(-) | ePhys EM | lj1 | wt | wt | no | n/a |
| RyR(-) | ePhys EM | wt | wt | e540 | n/a | n/a |
| CaV1(-) CaV2(-) | EM | lj1 | n582 | wt | no | no |
| CaV1(-) | EM | wt | n582 | wt | n/a | no |

in the nervous system (EG8409). The resulting construct *oxTi1047* was crossed into CaV1(-) / *egl-19(st556)* animals (RW3563), which rescued lethality but lacked expression in the nervous system.

To demonstrate that phenotypes in this EG9034/EG8409 were due to loss of nervous system function, we expressed the *egl-19* minigene under the neuron-specific P*snt-1* promoter and inserted the construct into the genome by miniMos (*Frøkjær-Jensen et al., 2014*). The resulting *oxTi1049* construct was crossed into the muscle-rescued CaV1(Δns) animals (EG9034) to generate EG9145.

Lethal double mutants of CaV2-RyR (genotype: *unc-2 (lj1); unc-68 (e540)*) and CaV2-CaV1 (genotype: *unc-2 (lj1); egl-19 (st556)*) were rescued by an extrachromosomal array expressing SNAP::CaV2/*unc-2* cDNA in a minimum set of acetylcholine head neurons, using a previously described truncated *unc-17* promoter, referred to as 'P*unc-17h*' (*Hammarlund et al., 2007*; *Topalidou et al., 2016*). The extrachromosomal array *oxEx2096* was generated in the *unc-2(lj1)* strain AQ130 and crossed to RyR / *unc-68(e540)* or CaV1 /*egl-19(st556) oxTi1047[Pset-18::egl-19b]* animals to generate double mutants. The resulting strains are lethal without the presence of *oxEx2096[Punc-17h::SNAP::unc-2]* and were used in electrophysiology experiments. *Table 1* contains a genotype summary of calcium channel mutants with allele designations of mutant alleles and rescues.

## Behavioral experiments

Animals were maintained under standard laboratory conditions. For behavioral experiments, 3–6 well-fed, young adult worms were transferred to a 10 cm containing standard NGM. Each assay was recorded for 5 min at 8 frames per second using the worm tracking software WormLab (2019.1.1, MBF Bioscience). The trajectory of each worm was collected using WormLab and imported into custom written R scripts for analysis. Worms that crawled out of the field of view during the first 3 min were discarded from analysis. Worms whose speed was lower than 100 µm/s were excluded as they may have been damaged during transfer, the number of worms that fell in this category were few and not different between groups. A reversal was defined as backwards locomotion that lasted more than 4 frames or 500ms.

## Generation of CaV2::HALO by CRISPR/cas9

CaV2 was tagged by CRISPR-mediated insertion of HALO coding DNA into the *unc-2* endogenous genomic locus. A DNA mix containing (1) PCR-generated DNA repair template that includes the HALO tag with an embedded *Cbr-unc-119(+)* cassette flanked by loxP sites and 33 bp homology arms to the cut site, (2) plasmid DNA that directs expression of Cas9 and an sgRNA (*Schwartz and Jorgensen, 2016*), and (3) an inducible negative selection plasmid directing expression of a histamine-gated

**Table 2.** Super-resolution alleles generated for this study.

| Allele | Gene | Common name | sgRNA | Repair template | Tag | Chr | Terminus |
|--------|------|-------------|-------|-----------------|-----|-----|----------|
| *ox672* | *unc-2* | CaV2 | pSAM429 (ACAGACCGCCAACCAACCGG) | pSAM593 | HALO | X | internal |
| *ox704* | *rimb-1* | RIMBP | TGGGTAAATCGATAAATCG | pSAM514 | SKY-S | III | c |
| *ox719* | *nrx-1* | Neurexin | TTTTCTTTGCCACCCCATTC | pSAM534 | SKY-S | V | c |
| *ox721* | *unc-68* | RyR | pSAM488 (gattagttagttccaagaaA) | pSAM593h | HALO | V | n |
| *ox727* | *ctn-1d* | α-Catulin | CATCCAATGTAATCGGC | pSAM598 | SKY-S | III | c |
| *ox728* | *egl-19* | CaV1 | CTTCTCATCCATTGCTC | pSAM604 | SNAP | IV | internal |
| *ox729* | *syd-1* | SYDE | GCACTGCGATTCCGAGACAT | pSAM545 | SKY-S | II | c |
| *ox730* | *syd-2* | α-Liprin | TTGCTGTAGCTCATatttct | pSAM549 | SKY-S | X | n |
| *ox747* | *elks-1* | ELKS/CAST | gagcagtacaatATGGCACC | pSAM550 | SKY-S | IV | n |
| *ox748* | *unc-13all* | UNC13all | gctttgaatccaacaaaaaa | pSAM613 | SKY-S | I | c |
| *ox803* | *magi-1* | MAGI | aagATGACCGACAAAACAGC | pSAM552 | SKY-S | IV | n |
| *ox814* | *unc-13b* | UNC13s | GGAACTGCAAGACTTGGCAC | pSAM684 | SKY-S | I | n |
| *ox802* | *unc-44* | Giant Ankyrin | GCTGTTGGTCGTGCTCCCGA | pSAM546 | SKY-S | IV | c |
| *ox708* | *unc-44* | Giant Ankyrin | GCTGTTGGTCGTGCTCCCGA | pSAM557 | SNAP | IV | c |

**Table 3.** Common names and nomenclature used in this study.

| Common name | Mammalian ortholog | C. elegans ortholog | Drosophila ortholog |
|---|---|---|---|
| CaV2 | CaV2.1 CaV2.2 CaV2.3 | UNC-2 | Cacophony |
| CaV1/L-type | CaV1.1 CaV1.2 CaV1.3 CaV1.4 | EGL-19 | DmCa1D |
| ELKS | ELKS/CAST | ELKS-1 | Bruchpilot |
| RyR | RYR1 RYR2 RYR3 | UNC-68 | dRyr |
| UNC13all | Munc13-1 ubMunc13-2 bMunc13-2 Munc13-3 | UNC-13L, UNC-13S | Unc13A Unc13B |
| UNC-13S | bMunc13-2 Munc13-3 | UNC-13S | UNC13B |
| RIMBP | RIMBP | RIMB-1 | Rbp |
| Veli/LIN7 | LIN7A | LIN-7 | Lin-7 |
| Giant Ankyrin | gAnkB | UNC-44L | AnkG |
| Neurexin/NRX | Neurexin 1 | NRX-1 | DNrx |
| α-Liprin | α-Liprin | SYD-2 | Liprin-α |
| SYDE | SYDE | SYD-1 | Syd-1 |
| α-Catulin | α-Catulin | CTN-1 | α-Cat |
| MAGI1 | MAGI1 | MAGI-1 | Magi |

chloride channel in neurons, pNP403 (*Pokala et al., 2014*) was injected into the gonads of young adult EG6207 *unc-119(ed3)* animals (*Maduro and Pilgrim, 1995*; *Schwartz and Jorgensen, 2016*; *Zhang et al., 2015*). Transgenic animals were selected for expression of *unc-119(+)*, and extrachromosomal-array bearing animals were selected against by addition of histamine to the media. The *loxP::Cbr-unc-119(+)::loxP* region of the insertion was excised by injecting pDD104[Peft-3::Cre] and identifying *unc-119(-)* animals (*Dickinson et al., 2013*). The modified locus introduces HALO-tag within an unconserved region in the second extracellular loop of CaV2 encoding UNC-2a. The resulting strain EG9823 (genotype: *unc-119*(ed3); *unc-2*(ox672[HALO])) was subsequently used to generate CRISPR-mediated insertions of Skylan-S tags.

## Generation of super-resolution Tags by CRISPR/cas9

Tags for other genes, including *egl-19, unc-68, elks-1, nrx-1, rimb-1, elks-1, syd-2, syd-1, magi-1, ctn-1, unc-13,* and *unc-13b* were constructed as previously described (*Schwartz and Jorgensen, 2016*). A single plasmid containing sgRNA and the repair template, composed of 57 bp homology arms and Skylan-S (*Zhang et al., 2015*) containing a *loxP::Cbr-unc-119(+)::loxP*, was appended by SapTrap plasmid assembly. Each assembled plasmid was mixed with plasmids to express Cas9 in the germline, and HisCl- in neurons, and injected into the gonads of young adult EG9823 animals. After selecting for *unc-119(+)* and selecting against extrachromosomal arrays by histamine application, animals were injected with pDD104[P*eft-3::Cre*], selected for excision of *loxP::Cbr-unc-119(+)::loxP*, and outcrossed once before analysis by super-resolution microscopy. *Table 2* contains a full list of superresolution alleles used in this study. *Table 3* contains a look up table for common nomenclatures for nematode, fly and mammalian homologs of protiens localized in this study.

## Strains

All strains were maintained at 22 °C on standard NGM media seeded with OP50.

| Name | Strain | Genotype |
|------|--------|----------|
| Wild-type | N2 | *wild-type* |
| CaV2(-) | AQ130 | *unc-2(lj1) X* |
| RyR(-) | CB540 | *unc-68(e540) V* |
| CaV1(-) balanced | RW3563 | *egl-19(st556) / unc-82(e1323) unc-24(e138am) IV* |
| EM Wildtype | EG5793 | *oxSi91[Punc-17::ChIEF::mCherry::unc-54UTR unc-119(+)] II; unc-119(ed9) III* |
| EM CaV2(-) | EG6584 | *oxSi91[Punc-17::ChIEF::mCherry::unc-54UTR unc-119(+)] II; unc-2(lj1) X* |
| EM CaV1(-) | EG6585 | *oxSi91[Punc-17::ChIEF::mCherry::unc-54UTR unc-119(+)] II; egl-19(n582) IV* |
| EM CaV2(-) CaV1(-) | EG6586 | *oxSi91[Punc-17::ChIEF::mCherry::unc-54UTR unc-119(+)] II; egl-19(n582) IV; unc-2(lj1) X.* |
| EM RyR(-) | EG6587 | *oxSi91[Punc-17::ChIEF::mCherry::unc-54UTR unc-119(+)] II; unc-68(e540) V* |
| CaV1 muscle rescue array | EG8409 | *egl-19(st556) IV; egl-19(oxEx2017[Pset-18::eGFP_egl-19b::let-858-utr; ccGFP])* |
| CaV1(Δns) RyR(-) | EG8827 | *egl-19(st556) IV; unc-68(e540) V; oxEx2017[Pset-18::eGFP::egl-19b::let-858-utr; Punc-122::GFP]* |
| CaV1(Δns) | EG9034 | *oxTi1047[Pset-18::egl-19b::let-858–3'utr] II; egl-19(st556)* |
| CaV2(Δnmj) RyR(-) | EG9405 | *unc-68(e540) V; unc-2(lj1) X; oxEx2097[Punc-17h::SNAP::unc-2]* |
| CaV2(Δnmj) CaV1(Δns) | EG9406 | *unc-2(lj1) oxTi1047[Pset-18::egl-19b::let-858 3'utr] II; egl-19(st556)IV; unc-2(lj1) X; oxEx2097[Punc-17h::SNAP::unc-2]* |
| RIMBP-SKYS CaV1-HALO (+ns rescue) Giant Ankryin-SNAP | EG9418 | *egl-19(st556) IV; ox704[Skylan-S::rimb-1] III; oxTi1047[Pset-18::egl-19b::let-858utr; HygroR(+)], oxTi1055[Psnt-1::HALO::egl-19b; NeoR(+)] II; unc-44(ox708[unc-44::snap]) IV* |
| CaV2-HALO Liprinα-SKYS | EG9425 | *unc-119(ed3) III; unc-2(ox672[HALO]), syd-2(ox715[Skylan-S(loxP::Cbr-unc-119(+)::loxP]) X* |
| RIMBP-SKYS CaV2-HALO CaV1-SNAP | EG9475 | *oxIs322[Cbr-unc-119(+) Pmyo-2::mCherry::histone Pmyo-3::mCherry::histone] II; unc-119(ed3) III; rimb-1(ox704[Skylan-S]) III; egl-19(ox728[snap]) IV; unc-2(ox672[HALO::]) X* |
| αCatulin-SKYS CaV1-SNAP CaV2-HALO | EG9476 | *ctn-1d(ox727[Skylan-S]) I; oxIs322[Cbr-unc-119(+) Pmyo-2::mCherry::histone Pmyo-3::mCherry::histone] II; unc-119(ed3) III; egl-19(ox728[SNAP]) IV; unc-2(ox672[HALO]) X* |
| NRX-SKYS CaV1-SNAP CaV2-HALO | EG9588 | *egl-19(ox728[SNAP]) IV; nrx-1(ox719[Skylan-S]) V; unc-2(ox672[HALO]) X* |
| ELKS-SKYS CaV1-SNAP CaV2-HALO | EG9617 | *elks-1(ox747[Skylan-S]), egl-19(ox728[SNAP]) IV; unc-2(ox672[HALO]) X* |
| ELKS-SKYS CaV1-SNAP RyR-HALO | EG9667 | *egl-19(ox728[SNAP]), elks-1(ox747[Skylan-S]) IV; unc-68(ox721[HALO]) V* |
| Giant Ankryin -SKYS CaV1-SNAP CaV2-HALO | EG9722 | *unc-2(ox672[HALO]) X; egl-19(ox728[SNAP]) IV; unc-44(ox802[Skylan-S]) IV* |
| UNC13all-SKYS CaV1-SNAP CaV2-HALO | EG9723 | *unc-2(ox672[HALO]) X; egl-19(ox728[SNAP]) IV; unc-13(ox748[Skylan-S]) I* |
| UNC13short-SKYS CaV1-SNAP CaV2-HALO | EG9782 | *unc-13(ox814[SKYLAN-S(loxP)]) I; unc-2(ox672[HALO]) X; egl-19(ox728[SNAP]) IV* |

| Name | Strain | Genotype |
|---|---|---|
| CaV2-HALO | EG9823 | *unc-2(ox672[HALO::unc-2]) X; unc –119(ed3) III* |
| RIMBP-SKYS<br>CaV1-HALO(+ns rescue)<br>Lin7-SNAP | EG10094 | *oxTi1055[Psnt-1::HALO::egl-19b; NeoR(+)] oxTi1047[Pset-18::egl-19b::let858utr; HygroR(+)] II; unc-119(ed3) rimb-1(ox704[Skylan-S]) III; egl-19(st556) IV; oxEx2223[Punc-129::lin-7::SNAPf-tag]* |
| SYDE-SKYS<br>CaV2-HALO | EG10095 | *syd-1(ox723[Skylan-S(loxP::Runc-119::loxP)]) II; unc-119(ed3) III; unc-2(ox672[HALO]) X* |
| MAGI-SKYS<br>CaV1-SNAP<br>CaV2-HALO | EG10096 | *unc-119(ed3) III; egl-1-19(ox728[snap]), magi-1(ox755[Skylan-S(loxP::Cbr-unc-119::loxP)]) IV; unc-2(ox672[HALO]) X* |

## Single molecule localization microscopy

Super-resolution images were recorded with a Vutara SR 352 (Bruker Nanosurfaces, Inc, Madison, WI) commercial microscope based on single molecule localization biplane technology (*Juette et al., 2008*; *Mlodzianoski et al., 2009*). First, 20–30 adult hermaphrodite *C. elegans* expressing HALO-tagged proteins *Encell et al., 2012*; *Mollwitz et al., 2012* were allowed to lay eggs for 6–8 hr. Adults were moved to a new plate or destroyed. ~2 days later, when L4s were abundant on the plate, but before young adults emerged, animals were washed off using M9 an briefly centrifuged and resuspended in M9 several times to remove bacteria. Dye solution was prepared by suspending 5 nmol of JF dye in 5 µl of fresh DMSO to make 1 mM solution of dye. Five µL of 1 mM solution was added to 95 µl of worms in M9. Animals were stained for 2 hr on an orbital shaker at room temperature in the dark in 50 µM of HTL-JF646, and 50 µM of STL-JF549cp, STL-JF549, or STL-JF549pa (Gift from Luke Lavis, Janelia Farms; *Grimm et al., 2017*; *Grimm et al., 2015*). Early super-resolution experiments were conducted with JF549-STL or PA-JF549-STL, we later found that a new cell permeable variant cp-JF549-STL improved labeling of channels. After 2 hr, 1 mL of M9 was added to the staining tubes, spun gently on a benchtop centrifuge and aspirated several times to remove dye. Animals were allowed to recover 12 hr at 15degC on agar seeded with OP50 bacteria. Molting is essential to remove non-specific staining of the cuticle. After they had molt, live intact animals were anesthetized in 25 mM NaN3 and regions of their dorsal cords that were positioned directly against the cover glass and away from the intestine. These were imaged with 640 nm excitation power of 10kW/cm2, or 549 nm excitation power of 5kW/cm2 Skylan-S was imaged by 488 nm excitation at 2kW/cm2, while photoactivated by 0.37 mW/cm2 405 nm light. Images were recorded using a 60 x/1.2 NA Olympus water immersion objective and Hamamatsu Flash4 V1 sCMOS, or 60 x/1.3 NA Silicon immersion objective and Orca Fusion BT SCMOS camera with gain set at 50 and frame rate at 50 Hz. Individual laser settings varied per animal to optimize blinking, but typically lasers were set below 12%. Example settings: 9% 646, 7% 549, 4% 488 with 2% 405 activation laser. A minimum of 1000 frames per fluorophore were recorded. Data was analyzed by the Vutara SRX software (version 7.0.0rc39). Single molecules were identified by their brightness frame by frame after removing the background. Identified molecules were localized in three dimensions by fitting the raw data in a 12x12-pixel region of interest centered around each particle in each plane with a 3D model function that was obtained from recorded bead data sets. Fit results were filtered by a density-based denoising algorithm to remove isolated particles. The experimentally achieved image resolution of 40 nm laterally (x,y) and 70 nm axially (in z) was determined by Fourier ring correlation. Localizations were rendered as 80 nm.

## SML analysis

Localization data was exclusively collected from the dorsal nerve cord, which contains axons and synapses but no neuronal soma. We performed a 3D reconstruction of *C. elegans* dorsal nerve cord to inform region of interest selection from fluorescent images. The orientation of dorsal cord synapses is predictable. Excitatory acetylcholine neurons and inhibitory GABA neurons synapse onto muscle arms (*Figure 3A*). These connections are near the edges of the cord bundle. Thus, the roll of the animal affects the orientation of the synapse; en face or axial.

For single molecule localization experiments, animals were rolled to ensure en face orientation of synapses. Synapses that were in focus and en face were analyzed. The average size of a synapse from the dorsal nerve cord is 579.7 nm (SEM +/-16 nm). Thus, super-resolution analysis regions of interest

were narrowed to localizations within 700 nm of the dense projection marker. Localization position data was flattened in the z-dimension due to chromatic aberrations. A script was used to calculate the center of each probe. To compare the distribution of probe A to probe B, an angle between the two clusters centers was calculated. The distribution distances were calculated by measuring the distance along the center-to-center axis from a probe B to the center of cluster A, and cluster B. Nearest neighbor analysis was done with knnsearch. The 95% confidence interval of these distance measurements is considered the diameter of the cluster. Distribution center and range or 'diameter' were reported as (mean, 95% CI). Proberuler available at https://github.com/bdmscience/proberuler (copy archived at *Mueller, 2022*).

## Counting calcium channels

CaV1 calculations. CaV1-SNAP was coupled to JF549cp. CaV1 channels are dispersed in the synapse but also often form small clusters, spatially distinct from CaV2 clusters (*Figure 10A*). To estimate the number of CaV1 channels, imaging conditions were optimized to maximize localizations from only CaV1 by imaging them first, and estimated their number using three approaches.

Total blinks. Each blink was assumed to be a channel (total CaV1 blinks $\bar{A}$=196 ± 37 channels). Since total blinks assumes every channel blinks only once, this method will represent an overcount of the true number of channels. Single channels emit multiple blinks (see next method), thus 200 channels likely represents the maximum number of channels.

Mean blinks per channel (~60 nm). Spatial information was used to determine how many blinks belong to the same channel – singlet blinks were assigned as a channel, and two or more blinks within 60 nm of each other were assumed to arise from a single channel via DBSCAN, based on a typical confidence interval (We observe 40 nm resolution by Fourier ring analysis, to be conservative a 60 nm arbitrary criterion was used, given that we only include blinks with better than 80 nm localization precision by Cramér-Rao lower bound; for context, a calcium channel with subunits is ~20 nm in diameter). A Poisson distribution was fit to the number of blinks per channel (m=2.7 blinks per channel). Dividing the total number of blinks in synapses by the mean number of blinks per channel produces an estimate of 79±10 CaV1 channels per synapse (*Figure 10B, C*).

Photon flux. Every blink has a photon count. The mode number of photons emitted in a blink was determined from *all blinks* at analyzed synapses (Mo = 575 photons / blink; $\bar{A}$=765 ± 11 photons/ blink; *Figure 10D*). Blinks were also clustered into single channels (as described above) and the number of *photons per 'channel'* was measured. Channels emitted a mean of 1996±489 photons because some channels blinked multiple times. The mode was 550 photons per channel (*Figure 10D*). The mode photon count for channels is similar to the mode photon count of single blinks indicating that most channels blink once. The total number of photons emitted by a synapse for the whole imaging session was divided by the mean number of photons per channel to estimate CaV1 channel count: (75±19 CaV1 channels; *Figure 10E*).

CaV2 calculations. CaV2-HALO was labelled with JF646. CaV2 channels are tightly localized into ~250 nm diameter clusters coincident with the dense projection. Each synapse contains just one density, but each density has several hundred CaV2 blinks. The number of CaV2 channels in individual boutons was estimated with the following approaches. Total blinks. CaV2 exhibited a mean 569±54 blinks / synapse. Again, as described above total blinks is an overestimate of the number of channels at a synapse.

Mean blinks or photon flux per channel. CaV2 blinks exhibited a mode of 900 photons. However, CaV2 localizations saturate the dense projection so single channels cannot be isolated. The cluster appears as a uniform mass, because the resolution and precision of our experiments are not adequate to spatially resolve single channels in densely labeled domains. To circumvent this limitation, the maximum number of channels in the density was calculated.

Cluster diameter – upper bound. An upper bound was estimated for the number of channels by mathematically fitting circles within a larger circle (20 nm channels in a 250 nm cluster) yields a maximum capacity of 120 channels at the dense projection.

Minimal overlap -- lower bound. The CaV2 clusters appears continuous without cavities, meaning all localizations overlap with neighboring channels. A mathematical fit of 60 nm circles within a 250 nm cluster results in at least 12 channels. Each channel would need to blink 50 times in 1 min. Organic dye blinking photophysics are not well characterized. However, this rate is approximately

10-fold higher than rates experimentally determined, as described below, suggesting there are many more channels.

Blinking rate. The blinking rate of dyes was used to estimate channel count. Although blinking rates of rhodamine dyes have not been determined, cyanine dyes in oxygen scavenging buffer can blink about six times in one minute depending on the imaging conditions (*Helmerich et al., 2022*). 569 blinks on average were observed from the CaV2 cluster in a one minute imaging interval. If rhodamine dyes are similar to cyanine dyes, then 101±16 CaV2s are present at a single synaptic bouton. Thus, 101 CaV2 channels is the best estimate given our limited knowledge of dye photophysics.

## Electrophysiology

All electrophysiological experiments were performed with young adult hermaphrodites. The animals were immobilized and dissected as previously described (*Liu et al., 2013*). In experiments to record minis mediated by both GABA and ACh receptors, the extracellular solution contained (in mM) NaCl 140, KCl 5, CaCl$_2$ 0.5, MgCl$_2$ 5, dextrose 11 and HEPES 5 (pH 7.2), and the pipette solution containing (in mM) KCl 120, KOH 20, Tris 5, CaCl$_2$ 0.25, MgCl$_2$ 4, sucrose 36, EGTA 5, and Na2ATP 4 (pH 7.2). In experiments to record minis mediated by only ACh receptors, the pipette solution was modified by reducing KCl to 6.8 mM and adding Kgluconate 112.2 mM. The holding voltage used in all the experiments was –60 mV. Dantrolene (D3996, TCI America) and nemadipine-A (SC-202727, Santa Cruz Biotechnology, Inc) stock solutions (10 mM) were made in DMSO. In each experiment, 1 μl of the stock solution was first thoroughly mixed with 50 μl of the bath solution in a small centrifuge tube, and then the entire volume of the mixed solution was added to the bath (total volume 1 ml) to reach the final concentration of 10 μM. After adding the dantrolene or nemadipine-A solution, the bath solution was pipetted up and down 5 times using the same pipette (set at 50 μl), and an incubation period of 5 min was allowed prior to starting the electrophysiological recording. The classic whole-cell configuration was used for voltage-clamp recordings with a Multiclamp 700B amplifier (Molecular Devices, Sunnyvale, CA, USA) and the Clampex software (version 10 or 11, Molecular Devices). Data were filtered at 2 kHz and sampled at 10 kHz. The frequency and amplitude of minis were quantified with ClampFit (version 11, Molecular Devices).

## Quantal modelling

A 3-term gaussian distribution was fit to the frequency distribution of amplitudes in 1 pA bins using MATLAB Curve Fitter (Mathworks, Natick, MA, USA). The terms were centered on 7+/-1 pA (nema alone) or 5+/-1 pA (dantrolene +nema) intervals which are the modes of the mini amplitudes, that is, a single quantum (*Del castillo and Katz, 1954*). The coefficient for the mean of the first gaussian curve was set to the mode amplitude (7 pA ±1). Every coefficient for subsequent gaussian terms were set to 7 pA ±1 intervals, but the other coefficients were not constrained and allowed to find a best fit. Area under each curve was calculated using MATLAB trapz.

## Flash and freeze electron microscopy

Electron microscopy was performed as previously described (*Watanabe et al., 2013*). Freezing was performed on a Leica EMpact2 (Leica, Wetzlar, Germany). To stimulate neurotransmission animals were exposed to blue (488 nm) LED light for 20ms and frozen 50ms later. Thirty-three nm serial sections were taken and imaged using a Hitachi H-7100 transmission electron microscope equipped with a Gatan Orius digital camera (Gatan, Pleasanton, CA). Micrographs were analyzed in ImageJ using a program for morphological analysis of synapses (*Watanabe et al., 2020*). Scripts available at: https://github.com/shigekiwatanabe/SynapsEM (copy archived at *Watanabe, 2022*).

## Dorsal nerve cord reconstruction

Serial sections were cut at 100 nm and imaged using JEOL JEM-1400 (JEOL, Peabody, MA) then annotated and assembled using TrackEM2 in FIJI (*Cardona et al., 2012*). Specifically, a wireframe was fit through each process that was suspected to be in the previous micrograph. Then an outline of the plasma membrane of each process was drawn. We analyzed several criteria to more specifically determine the specific process name and type: the morphology of each process and compared to previously published data (*White et al., 1986*), and the number of synapses. These data allow us to determine the identity of a process with some certainty.

## Whole genome sequencing of *unc-68(e504)*

CB504 animals were grown on HB101 and pelleted. DNA was extracted with a Qiagen DNeasy Blood and Tissue kit (Qiagen #69504) and eluted in EB buffer. Sequencing libraries were prepared using a Nextera XT DNA Library preparation kit (Illumina) and sequenced on an Illumina NovaSeq using paired 150-base reads. Reads were aligned to the *C. elegans* genome (WS276) with bwa (*Li and Durbin, 2009*) and processed with Samtools (*Li et al., 2009*). Aligned reads were base-called with GATK (*McKenna et al., 2010*) and mutations were annotated with SnpEff (*Cingolani et al., 2012*). Nextera library preparation and sequencing was performed by the Huntsman Cancer Institute High Throughput Genomics core facility. The VCF file describing *unc-68(e540)* can be found in *Source data 1*. Research reported in this publication utilized the High-Throughput Genomics and Bioinformatic Analysis Shared Resource at Huntsman Cancer Institute at the University of Utah and was supported by the National Cancer Institute of the National Institutes of Health under Award Number P30CA042014. The content is solely the responsibility of the authors and does not necessarily represent the official views of the NIH.

tttttttttcagGAAAAATCAT (wild-type)
tttttttttcaaGAAAAATCAT (e540)

## Acknowledgements

We thank Lexy von Diezmann for development of Proberuler. We thank Luke Lavis for providing all Janelia Fluor dyes (JF Dyes). Wayne Davis suggested using the early expressing *set-18* promoter to rescue CaV1. We thank the University of Utah Fluorescence Imaging Core for instrumentation, and the *Caenorhabditis* Genetics Center (CGC) for maintaining and distributing strains to the *C. elegans* community. We thank Patrick McEachern, Matthew Rich, Jessica Vincent and M Wayne Davis for their critical reviews of this manuscript. EMJ is an investigator at Howard Hughes Medical Institute. We thank Dominique Glauser the gift of *unc-68(syb216)*.

## Additional information

### Funding

| Funder | Grant reference number | Author |
| --- | --- | --- |
| National Science Foundation | NeuroNex 2014862 | Erik M Jorgensen |
| National Institutes of Health | R01 NS034307 | Erik M Jorgensen |
| National Institutes of Health | R01 MH085927 | Zhao-Wen Wang |
| National Institutes of Health | R01 NS109388 | Zhao-Wen Wang |
| National Institutes of Health | R01 NS094421 | Andres Villu Maricq |
| National Institutes of Health | F31 NS084826 | Sean A Merrill |

The funders had no role in study design, data collection and interpretation, or the decision to submit the work for publication.

### Author contributions

Brian D Mueller, Data curation, Formal analysis, Investigation, Methodology, Project administration, Resources, Software, Supervision, Validation, Visualization, Analyzed electrophysiology data; Performed SMLM; Designed Proberuler; Analyzed SMLM data; Performed genetic crosses; Cloned Plasmids; Generated Transgenic Animals; Annotated serial reconstruction; Sean A Merrill, Conceptualization, Data curation, Formal analysis, Investigation, Methodology, Project administration, Resources,

Supervision, Validation, Visualization, Performed SMLM; Analyzed SMLM; Performed genetic crosses; Cloned plasmids; Generated Transgenic animals; Generated all CRISPR tags used in this study; Shigeki Watanabe, Conceptualization, Formal analysis, Investigation, Validation, Resources, Software, Supervision, Visualization, Project administration, Performed and analyzed time resolved electron microscopy; Ping Liu, Formal analysis, Validation, Resources, Software, Project administration, Performed electrophysiology; Analyzed electrophysiology; Longgang Niu, Formal analysis, Validation, Resources, Software, Performed electrophysiology; Analyzed electrophysiology; Anish Singh, Validation, Software, Annotated serial reconstruction; Performed serial reconstruction EM; Pablo Maldonado-Catala, Formal analysis, Validation, Resources, Software, Supervision, Performed behavioral recordings and analysis; Alex Cherry, Data curation, Software, Cloned Plasmids; Performed genetic crosses; Matthew S Rich, Formal analysis, Validation, Software, Performed WGS and analysis; Malan Silva, Formal analysis, Validation, Software, Annotated serial reconstruction; Performed serial reconstruction EM; Andres Villu Maricq, Performed electrophysiology. Analyzed electrophysiology., Annotated serial reconstruction. Performed serial reconstruction EM., Methodology; Zhao-Wen Wang, Performed electrophysiology. Analyzed electrophysiology., Software, Methodology, Project administration; Erik M Jorgensen, Conceptualization, Data curation, Performed electrophysiology. Analyzed electrophysiology., Annotated serial reconstruction. Performed serial reconstruction EM., Supervision, Methodology, Project administration

## Author ORCIDs
Brian D Mueller ⓘ http://orcid.org/0000-0002-6525-7101
Shigeki Watanabe ⓘ http://orcid.org/0000-0001-7580-8141
Longgang Niu ⓘ http://orcid.org/0000-0001-7209-7436
Zhao-Wen Wang ⓘ http://orcid.org/0000-0003-3574-8556
Erik M Jorgensen ⓘ http://orcid.org/0000-0002-2978-8028

## Decision letter and Author response
Decision letter https://doi.org/10.7554/eLife.81407.sa1
Author response https://doi.org/10.7554/eLife.81407.sa2

---

# Additional files

## Supplementary files
- MDAR checklist
- Source data 1. VCF file of CB540 *unc-68(e540)*.

## Data availability
All data generated and analyzed during this study have been included as supporting files. Source data files have been provided. Github links to scripts used for analysis are noted in the methods.

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
