## [Editor Report]

Using an elegant combination of cutting-edge techniques, the authors show that in the neuromuscular junction of the nematode *C. elegans* two different classes of voltage-activated calcium channels differentially trigger exocytosis of distinct pools of synaptic vesicles, one docked to the active zone and a second one localized more distant from the active zone. These findings provide fascinating and novel insights into a classical problem of presynaptic physiology.

---

## [Decision Letter]

**Decision letter after peer review:**

Thank you for submitting your article "CaV1 and CaV2 calcium channels mediate the release of distinct pools of synaptic vesicles" for consideration by *eLife*. Your article has been reviewed by 3 peer reviewers, including Reinhard Jahn as the Reviewing Editor and Reviewer #1, and the evaluation has been overseen by Kenton Swartz as the Senior Editor. The following individuals involved in the review of your submission have agreed to reveal their identity: Joshua M Kaplan (Reviewer #2); Kang Shen (Reviewer #3).

All three reviewers agree that the manuscript addresses a highly interesting question, that the data are generally convincing, and recommend publication after appropriate revision. Whereas reviewers 1 and 3 did not have any significant concerns, Reviewer 2 has given very detailed feedback to the authors that they are encouraged to consider during revision. After internal discussion, the referees agree that the following points should be addressed:

Essential revisions:

1) Most importantly, the referees are concerned about possible indirect (long-term) effects of the RYR inactivation in the mutants. They, therefore, recommend adding a series of experiments in which acute pharmacological blockade of calcium release from the ER using appropriate inhibitors (such as CPA) is used, These recordings should be done in all conditions where the RYR mutations were used (i.e. CPA alone, CPA + CaV1 antagonist, and CPA in CaV2 mutants). These additional experiments would considerably strengthen the main conclusions.

2) The reviewers also agree that it needs to be clarified whether the recording conditions for the mini GAB Aergic and cholinergic minis can be clearly differentiated, considering the high chloride concentrations in the recording electrode. This may just be a clerical error in the methods, but it needs to be ensured that only cholinergic EPSCs are recorded.

*Reviewer #1 (Recommendations for the authors):*

As stated, the experiments appear to be of high quality, and I have no suggestions for improvement.

*Reviewer #2 (Recommendations for the authors):*

I have a few concerns that hopefully can be addressed fairly easily.

1) Are the ephys recordings in Figure 2 and its supplement a mixture of ACh and GABA synapses? The methods suggest that a high Cl^-^ pipet solution was used in these recordings, which would make GABA currents excitatory. Is this correct? If this is the case, any changes in mini rates or amplitudes could reflect changes in the balance of ACh and GABA transmission. It would be much better to isolate the ACh synaptic currents (i.e. using a low chloride internal solution). I don't see how quantal modeling (Figure 2F and G) is possible if this is a mixture of ACh and GABA minis.

2) Many of the results shown in Figure 2 replicate findings in a prior study (Tong et al., Neuron 2017), including mini rates in WT+Nemad, unc-2, unc-2+Nemad, egl-19(ns), and egl-19(ns)+Nemad. Tong et al. also concluded that half the minis are mediated by UNC-2 and the other half by EGL-19. The text should be revised to indicate which findings are replications of prior studies, and which are new. You should also revise statements in the text claiming that this is the first study to show that UNC-2 and EGL-19 both promote release at this synapse.

3) The contribution of RYR receptors to release is inferred from the analysis of unc-68(e540) mutants. Consequently, secondary or compensatory effects of the RYR mutations could contribute to the effects reported here. For example, a prior study suggests that the ER network in muscles is fragmented in unc-68 mutants (Maryon, JCS 1998). In addition, long-term changes in ER calcium homeostasis in unc-68 mutants could lead to many downstream physiological defects (including effects of calcium stores levels on surface ion channels). It would strengthen the conclusions if the ephys recordings done in unc-68 mutants (Figure 2, and Figure 2 extended data) could be repeated in animals acutely treated with SERCA inhibitors (e.g. CPA or thapsigargin). Also, is e540 a null allele? I cannot find a description of the molecular defect in this allele.

4) The hypomorphic allele, egl-19(n582), used to study CaV1's contribution to SV fusion in the EM flash-freeze experiments (Figure 3), is not a good choice. A prior study showed that EGL-19 current amplitudes are not significantly reduced in n582 mutants (Gao, PNAS 2011). This likely explains why the CaV1 defects (Figure 3F) look weaker than those in the RYR mutant defects (Figure 3G). It is surprising that n582 has a stronger defect in the 33 nm window than RYR deficient mutants, which I don't understand. Given these concerns, do the n582 flash-freeze data strengthen any of the conclusions? If these results are retained, the authors need to clearly explain that the n582 mutation may have a fairly mild CaV1 defect. The egl-19(ad1006) mutation has a much stronger defect in EGL-19 current and would be a better choice for these experiments. Has this analysis been done in the CaV1(ns) mutants?

5) Could the calcium dependence of SV priming/docking contribute to the flash-freeze defects seen in Figure 3? The changes in docked SVs following optogenetic stimulation are interpreted as resulting from changes in the stimulus-evoked calcium transient. An alternative explanation is that the long-term effects of calcium channel mutants on SV docking and priming could contribute to altered SV fusion rates.

6) Is baseline SV docking increased in unc-2 mutants (0-66 nm window, Figure 3E). Is that effect significant? This could be a consequence of the 50% decrease in mini rate in this mutant, resulting in more unfused docked SVs.

7) Is baseline SV docking altered in the unc-68 mutants (Figure 3G)? To my eye, it looks like baseline docking is decreased in the distal AZ (200-600 nm). Not sure I understand this because unc-68 and unc-2 have similar decreases in mini rate. This (and the preceding comment #6) could indicate that the spatial pattern of SV priming/docking is altered by the CaV2 and RYR mutations

8) Are lateral SV fusions (33-166 nm) significantly decreased in unc-2 mutants (Figure 3E)? This would suggest that CaV2 does promote lateral SV fusions, potentially by activating RYR channels.

9) Are distal SV fusions (198-400 nm) exaggerated in unc-2 mutants (Figure 3E)? If so, how do you explain this?

10) Please provide a more complete description of the super-res imaging methods (Figures 4-8). I'm not sure that I fully understand these data. The images shown are apparently derived from single-molecule localizations. So, the macroscopic puncta shown in these figures result from superimposing many localizations? If so, please indicate how many localizations are shown for each image. Can you estimate the absolute number of molecules present in each punctum? This would be interesting, especially for CaV2, CaV1, RYR, and UNC-13. Is there any substructure evident in these signals, e.g. nanoclusters of CaV2, CaV1, or RYR. If so, it would be very interesting to provide details about these nanoclusters (size, spatial frequency, number of molecules in each).

11) Do you have data showing that the tagged UNC-2, EGL-19, UNC-68, and UNC-13 alleles have no effect on function?

---

## [Author Response]

Essential revisions:1) Most importantly, the referees are concerned about possible indirect (long-term) effects of the RYR inactivation in the mutants. They, therefore, recommend adding a series of experiments in which acute pharmacological blockade of calcium release from the ER using appropriate inhibitors (such as CPA) is used, These recordings should be done in all conditions where the RYR mutations were used (i.e. CPA alone, CPA + CaV1 antagonist, and CPA in CaV2 mutants). These additional experiments would considerably strengthen the main conclusions.

The reviewers are concerned that the reduction in miniature frequencies observed in *unc-68(-)* animals are due to developmental defects rather than defects in exocytosis due to a loss of calcium influx. To address this concern, we performed a new set of electrophysiology experiments with the ryanodine receptor inhibitor dantrolene. We chose dantrolene, instead of the SERCA inhibitors cyclopiazonic acid (CPA) and thapsigargin, to analyze direct inhibition of the ryanodine receptor. In addition, we recorded from *unc-68(syb216)* animals have a deletion of the neuron-specific isoform of the ryanodine receptor to demonstrate that UNC-68 acts presynaptically to regulate minis *(*Marques et al. 2020). These new experiments were performed using a low-chloride pipette solution at a holding voltage equal to the chloride equilibrium potential to isolate minis mediated by ACh transmission, which addresses a major consideration from reviewer 2.

Mini Frequencies. Dantrolene reduced the frequency of minis in wild-type worms by about 50%. The inhibitory effect of dantrolene was not exacerbated by the CaV1 channel blocker nemadipine, which by itself inhibited the frequency of minis. Miniature current frequencies in the *unc-68(syb216)* strain were reduced compared with the wild type. However, the reduction in minis was less than that caused by dantrolene. This discrepancy is consistent with residual expression of UNC-68 in neurons from the muscle promoter (about 12% of mRNA transcripts in neurons are the *unc-68* muscle isoform) (Marques et al. 2020; Ma et al. 2016). These results (shown in Figure 2C) support our conclusion that presynaptic ryanodine receptors act together with Ca_V_1 to regulate synaptic vesicle exocytosis at the *C. elegans* neuromuscular junction.

p7 “To confirm that the reduction of minis observed in the *unc-68* null mutant was due to an acute loss of RyR function rather than a developmental defect, we blocked RyR by applying dantrolene (10 µM), a specific RyR inhibitor (Song, Karl, Ackerman, & Hotchkiss, 1993; Xu, Tavernarakis, & Driscoll, 2001), either alone or in combination with the CaV1 blocker nemadipine. The recordings were performed using a low-chloride pipette solution at a holding voltage equal to the chloride equilibrium potential so that only minis mediated by acetylcholine were detected. The frequency of minis was reduced in all treatments compared to the wild type (Figure 2C; wild type 21.7 ± 1.0 mini/s; dantrolene alone 12.36 +/- 0.6 mini/s; nemadipine alone 15.8 +/- 0.7 mini/s; dantrolene plus nemadipine 12.2 +/- 0.9 mini/s). Furthermore, nemadipine did not exacerbate the inhibitory effect of dantrolene on minis. Again, CaV1 is reliant on RyR for neurotransmission.

To confirm that RyR is acting presynaptically we analyzed strains rescued in neurons or in muscle. Previously, we rescued minis in a null mutant by expressing wild-type *unc-68* in neurons but not muscle cells, suggesting that RyRs regulate minis by acting presynaptically (Liu et al., 2005). Expression from *unc-68* is driven by an upstream muscle promoter and a downstream neuronal promoter (Chen et al., 2017; Marques et al., 2020). To confirm the presynaptic function of RyRs, we recorded minis from the RyR(Δns) mutant *unc-68(syb216)* (Figure 2C). The frequency of minis was significantly reduced compared to the wild type (wild type 21.7 ± 1.0 mini/s; RyR(Δns) 15.9 ± 0.6 mini/s; p=0.001). The reduction in minis in RyR(Δns) supports the conclusion that RyR is required presynaptically for normal levels of synaptic vesicle fusion. Mini frequency in RyR(Δns) was slightly higher than pharmacological block by dantrolene (12.36 +/- 0.6 mini/s; p=0.002). This result is consistent with the observation ~10% of RyR transcripts in neurons are expressed from the “muscle” promoter in this strain (Marques et al., 2020).”

Amplitude. Pharmacological block of CaV1 with nemadipine or RyR with dantrolene significantly reduces mini amplitude. The effect of dantrolene and nemadipine is no worse than dantrolene alone, and a similar decrease in amplitude is observed in *unc-68(syb216)*. We conclude that acute loss the ryanodine receptor reduces current amplitudes. The most likely cause is due to a loss of multivesicular release driven by calcium from internal stores. Although it is possible that the effect could be due to rapid changes in the postsynaptic receptor field, the modal mini amplitude is unchanged, which suggests a presynaptic loss of multivesicular fusion events. These data can be found in Figure 2E.

p7-8, “Using recording conditions in which only acetylcholine currents are detected, the modal value of miniature currents was similar in all strains (wild type 8pA, CaV1 block nemadipine 7pA, RyR block dantrolene 5pA, CaV1+RyR block nemadipine+ dantrolene 6pA, RyR(Δns) 6pA), suggesting that the receptor field is similar for most single vesicle fusions. Simultaneous fusion of multiple vesicles – multiquantal release – will increase the mean current amplitude. The mean current amplitude from acetylcholine release was significantly reduced by pharmacological or genetic block of RyR in neurons (wild type:12.4 ±0.6 pA; dantrolene: 8.0 ±0.3 pA; RyR(Δns): 9.5 ±0.4 pA) (Figure 2D,E). The presence of large current events was not reduced by mutation of CaV2 (Figure 2 Supplement D-G). Block of CaV1 by nemadipine caused a decrease in the mean current amplitude (wild type:12.4 ±0.6 pA; nemadipine: 10.1 ±0.6 pA; nemadipine + dantrolene: 8.2 ±0.24 pA), suggesting that CaV1 contributes to multiquantal release. “

Quanta. We repeated the multiquantal release analysis on the wild-type worms treated with nemadipine alone or dantrolene and nemadipine together. We found that RyR inhibition resulted in the loss of multiquantal events greater than 2 quanta. We conclude that CaV1 and RyR, as well as CaV2 and RyR, are interacting to promote multiquantal release in acetylcholine neurons. These data can be found in Figure 2F-G.

Figure 2 and the figure 2 legend have been updated with these data.

Behavior. We performed behavioral analysis of *unc-68(syb216)* ‘RyR(Δns)’. Overall, these animals appear very similar to the wild type, except that they are slower (Figure 1 supplement). It is important to note this is not a complete knockout in the nervous system, remaining RyR isoforms likely rescue some of the behavior.

p5, " If CaV1 and RyR are functioning in the same pathway then worms lacking expression of RyR in the nervous system should phenotypically mimic CaV1(Δns). *unc68(syb216)* animals lack the neuronal-specific isoform of RyR, but express ~10% levels of RyR in neurons from the muscle promoter (Ma et al., 2016; Marques et al., 2020). We will refer to this strain as RyR(Δns) for simplicity. Like CaV1(Δns) animals, RyR(Δns) animals are significantly slower than the wild type; however, they do not exhibit the same increase in reversal frequency (Figure 1 Supplement A-I).”

We have added Figure 1 Supplement with RyR(Δns) mutant behavior compared to WT and added a figure legend to Figure 1 Supplement.

2) The reviewers also agree that it needs to be clarified whether the recording conditions for the mini GAB Aergic and cholinergic minis can be clearly differentiated, considering the high chloride concentrations in the recording electrode. This may just be a clerical error in the methods, but it needs to be ensured that only cholinergic EPSCs are recorded.

The reviewer concerns were valid; the previous recordings were performed under conditions where both ACh and GABA minis appeared as inward current events. To avoid potential complications of our multiquantal analyses by GABA events, we performed a new set of experiments under conditions in which only ACh minis were detected. Acute block of the ryanodine receptor leads to a loss of multiquantal events, and these multiquantal events can be stimulated by calcium influx from either CaV1 or CaV2.

p5, " Body muscles were voltage-clamped at a holding potential of -60mV, and miniature postsynaptic currents recorded under high chloride internal pipette conditions, in which GABA and ACh release generates inward currents."

p7, " Using recording conditions in which only acetylcholine currents are detected, the modal value of miniature currents was similar in all strains (wild type 8pA, CaV1 block nemadipine 7pA, RyR block dantrolene 5pA, CaV1+RyR block nemadipine+ dantrolene 6pA, RyR(Δns) 6pA), suggesting that the receptor field is similar for most single vesicle fusions.” p8, " CaV2 is also functionally coupled to RyR to drive multiquantal release in acetylcholine motor neurons.”

Reviewer #2 (Recommendations for the authors):I have a few concerns that hopefully can be addressed fairly easily.1) Are the ephys recordings in Figure 2 and its supplement a mixture of ACh and GABA synapses? The methods suggest that a high Cl^-^ pipet solution was used in these recordings, which would make GABA currents excitatory. Is this correct? If this is the case, any changes in mini rates or amplitudes could reflect changes in the balance of ACh and GABA transmission. It would be much better to isolate the ACh synaptic currents (i.e. using a low chloride internal solution). I don't see how quantal modeling (Figure 2F and G) is possible if this is a mixture of ACh and GABA minis.

High chloride**.** The reviewer is correct, and we thank the reviewer for the close read. In the original set of experiments we used a high internal chloride solution, in which GABA release produces inward currents. This confounds the analysis we performed on multiquantal release because GABA and acetylcholine minis will contribute to mini amplitudes. We have repeated the pharmacological experiments under low chloride internal pipette conditions. These data can be found in Figure 2C-G.

We have also added the following to the manuscript.

p5, " Body muscles were voltage-clamped at a holding potential of -60mV, and miniature postsynaptic currents recorded under high chloride internal pipette conditions, in which GABA and ACh release generates inward currents."

p7, " Using recording conditions in which only acetylcholine currents are detected, the modal value of miniature currents was similar in all strains (wild type 8pA, CaV1 block nemadipine 7pA, RyR block dantrolene 5pA, CaV1+RyR block nemadipine+ dantrolene 6pA, RyR(Δns) 6pA), suggesting that the receptor field is similar for most single vesicle fusions.” p8, " CaV2 is also functionally coupled to RyR to drive multiquantal release in acetylcholine motor neurons.”

2) Many of the results shown in Figure 2 replicate findings in a prior study (Tong et al., Neuron 2017), including mini rates in WT+Nemad, unc-2, unc-2+Nemad, egl-19(ns), and egl-19(ns)+Nemad. Tong et al. also concluded that half the minis are mediated by UNC-2 and the other half by EGL-19. The text should be revised to indicate which findings are replications of prior studies, and which are new. You should also revise statements in the text claiming that this is the first study to show that UNC-2 and EGL-19 both promote release at this synapse.

We in fact cited Tong et al. 2017 in the Introduction for observing that both CaV2 and CaV1 contributed to transmission at neuromuscular junctions. However, we did not state that each contributed half (which is an important point). In addition, we did not cite the previous publication in the Results section when we describe the results of our physiological experiments. We apologize for the oversight. We have added citations to these prior findings to the text.

p3, " Physiological studies suggest CaV1 can also contribute to neurotransmission; CaV1 channel blockers reduce tonic minis *by half* (Tong et al., 2017)." p4, Here, we demonstrate in *C. elegans* that two different classes of voltage-gated calcium channels, CaV2 (UNC-2) and CaV1 (EGL-19) mediate the release of two physiologically distinct pools of synaptic vesicles as described in an earlier study (Tong et al. 2017). p6, We conclude that all vesicle fusion at neuromuscular junctions relies on CaV1 and CaV2, each contributing about half of the minis, in agreement with an earlier study (Tong et al., 2017).

3) The contribution of RYR receptors to release is inferred from the analysis of unc-68(e540) mutants. Consequently, secondary or compensatory effects of the RYR mutations could contribute to the effects reported here. For example, a prior study suggests that the ER network in muscles is fragmented in unc-68 mutants (Maryon, JCS 1998). In addition, long-term changes in ER calcium homeostasis in unc-68 mutants could lead to many downstream physiological defects (including effects of calcium stores levels on surface ion channels). It would strengthen the conclusions if the ephys recordings done in unc-68 mutants (Figure 2, and Figure 2 extended data) could be repeated in animals acutely treated with SERCA inhibitors (e.g. CPA or thapsigargin). Also, is e540 a null allele? I cannot find a description of the molecular defect in this allele.

The reviewer notes that mutations in the ryanodine receptor could result in developmental defects or cell-nonautonomous defects in synaptic function. We agree with the reviewer and have repeated a group of experiments using dantrolene to acutely inhibit the ryanodine receptor for physiology experiments. In addition, we have recorded from a mutant with intact expression of the ryanodine receptor in the muscle, but reduced expression in the nervous system *unc-68(syb216)*. This strain lacks the neuron promotor and the neuron-specific exon 1.2; this isoform is expressed 7-fold higher than exon 1.1 in neurons (Marques et al. 2020; Ma et al. 2016). The pharmacological disruption and the neuron-specific reduction of RyR function is consistent with the defects observed in the null mutant in our previous results.

p7, “To confirm that the reduction of minis observed in the *unc-68* null mutant was due to an acute loss of RyR function rather than a developmental defect, we blocked RyR by applying dantrolene (10 µM), a specific RyR inhibitor (Song, Karl, Ackerman, & Hotchkiss, 1993; Xu, Tavernarakis, & Driscoll, 2001), either alone or in combination with the CaV1 blocker nemadipine. The recordings were performed using a low-chloride pipette solution at a holding voltage equal to the chloride equilibrium potential so that only minis mediated by acetylcholine were detected. The frequency of minis was reduced in all treatments compared to the wild type (Figure 2C; wild type 21.7 ± 1.0 mini/s; dantrolene alone 12.36 +/- 0.6 mini/s; nemadipine alone 15.8 +/- 0.7 mini/s; dantrolene plus nemadipine 12.2 +/- 0.9 mini/s). Furthermore, nemadipine did not exacerbate the inhibitory effect of dantrolene on minis. Again, CaV1 is reliant on RyR for neurotransmission.

To confirm that RyR is acting presynaptically we analyzed strains rescued in neurons or in muscle. Previously, we rescued minis in a null mutant by expressing wild-type *unc-68* in neurons but not muscle cells, suggesting that RyRs regulate minis by acting presynaptically (Liu et al., 2005). Expression from *unc-68* is driven by an upstream muscle promoter and a downstream neuronal promoter (Chen et al., 2017; Marques et al., 2020). To confirm the presynaptic function of RyRs, we recorded minis from the RyR(Δns) mutant *unc-68(syb216)* (Figure 2C). The frequency of minis was significantly reduced compared to the wild type (wild type 21.7 ± 1.0 mini/s; RyR(Δns) 15.9 ± 0.6 mini/s; p=0.001). The reduction in minis in RyR(Δns) supports the conclusion that RyR is required presynaptically for normal levels of synaptic vesicle fusion. Mini frequency in RyR(Δns) was slightly higher than pharmacological block by dantrolene (12.36 +/- 0.6 mini/s; p=0.002). This result is consistent with the observation ~10% of RyR transcripts in neurons are expressed from the “muscle” promoter in this strain (Marques et al., 2020).”

*unc-68(e540)*. Although *unc-68(e540)* is widely regarded as a putative null, the mutation was ambiguously identified in the literature (Sakube et al. 1997). We sequenced the genome of *unc68(e540)* mutants, and have reported the mutation in WormBase. *e540* is a G>A splice acceptor mutation near the middle of the protein:

ttttttttcagGAAAAATCAT (wild-type) ttttttttcaaGAAAAATCAT (e540)

p5, " *unc-2(lj1)* is a large deletion and frame shift, and *unc-68(e540)* is a G>A splice acceptor mutation near the middle of the protein, likely causing a null phenotype (Sakube, Ando, & Kagawa, 1997; this paper)."

4) The hypomorphic allele, egl-19(n582), used to study CaV1's contribution to SV fusion in the EM flash-freeze experiments (Figure 3), is not a good choice. A prior study showed that EGL-19 current amplitudes are not significantly reduced in n582 mutants (Gao, PNAS 2011). This likely explains why the CaV1 defects (Figure 3F) look weaker than those in the RYR mutant defects (Figure 3G). It is surprising that n582 has a stronger defect in the 33 nm window than RYR deficient mutants, which I don't understand. Given these concerns, do the n582 flash-freeze data strengthen any of the conclusions? If these results are retained, the authors need to clearly explain that the n582 mutation may have a fairly mild CaV1 defect. The egl-19(ad1006) mutation has a much stronger defect in EGL-19 current and would be a better choice for these experiments. Has this analysis been done in the CaV1(ns) mutants?

As noted, the *egl-19(n582)* is a weak mutation. Unfortunately, there was no other option: The complete absence of function of CaV1 in the nervous system causes lethality in the double mutant CaV1(Δns) CaV2(-) (Figure 1A). These data mean that CaV1 and CaV2 are partially redundant in the nervous system for viability, and specific loss in the nervous system of both channels is synthetic lethal. Using an allele that is viable in the double mutant is the best we can do in this experiment.

However, the goal of this experiment was to determine if loss of both CaV1 and CaV2 causes reduced fusion at both the dense projection and distal sites. In fact, in the CaV1 CaV2 double mutant there is no significant fusion remaining at the dense projection or at lateral sites, supporting the conclusion that these channels are essential for fusions occurring at the dense projection and at lateral sites. The fraction of vesicles that fuse at lateral sites in the *egl19(n582)* mutant is less than in the wild-type (chi-square P<0.001), but is not reduced to zero (Figure 3 supplement D). However, the ultrastructural analysis could only be performed on a living animal. We now cite the Table in Figure 1A to clarify why we used a weak allele in our double mutant.

p9, "Complete loss of both CaV2 and CaV1 function in the nervous system is lethal (Figure 1A).

Therefore, to assay mutation of both channels in the nervous system we used a weak allele of CaV1; the hypomorph *egl-19(n582)* is viable in double mutants with *unc-2(lj1)*."

5) Could the calcium dependence of SV priming/docking contribute to the flash-freeze defects seen in Figure 3? The changes in docked SVs following optogenetic stimulation are interpreted as resulting from changes in the stimulus-evoked calcium transient. An alternative explanation is that the long-term effects of calcium channel mutants on SV docking and priming could contribute to altered SV fusion rates.

The reviewer is concerned that calcium channel mutants could have developmental effects on synaptic anatomy or function. In the resubmission, we used acute pharmacological blocks of calcium channels and found drug block of channels phenocopied the mutants.

6) Is baseline SV docking increased in unc-2 mutants (0-66 nm window, Figure 3E). Is that effect significant? This could be a consequence of the 50% decrease in mini rate in this mutant, resulting in more unfused docked SVs.

The next four considerations relate to the role of calcium channels in synaptic vesicle fusion at different spatial areas of the active zone as analyzed by electron microscopy. We define three zones in this resubmission: proximal to the dense projection (0-33nm), intermediate active zone (33-165nm) and the distal active zone (165-594nm).

The reviewer suggests that there is an increase in docked vesicles in the proximal zone adjacent to the dense projection in the CaV2 channel mutant. There is an increase in docked vesicles in the 0-33 nm bin in the *unc-2(lj1)* compared to the wild type, but the increase does not reach significance (Figure 3 supplement A; Brown-Forsythe and Welch's ANOVA and T3 Dunnett’s). As suggested by the reviewer, it is highly likely that synaptic vesicles accumulate adjacent to the dense projection due to loss of tonic fusion at these sites.

Unfortunately, reconstructing synapses from serial electron micrographs is extremely demanding. Because analysis was performed on approximately a dozen reconstructed synapses, we lack the statistical power to conclude the changes in distribution are significant. We’ve added this information in Figure 3 supplement A.

p9, " To more finely partition the fusion domains for which each channel, we binned micrographs into three zones: adjacent to the dense projection (0-33 nm), intermediate active zone (33-165 nm), and lateral active zone (165-594 nm). Baseline docking was increased in the 0-33 nm bin in the CaV2(-) mutants, this trend is consistent with decreased tonic fusion adjacent to the dense projection (Figure 3 Supplement A), and release probability of vesicles is reduced adjacent to the dense projection in CaV2(-) mutants (Figure 3 Supplement D).”

7) Is baseline SV docking altered in the unc-68 mutants (Figure 3G)? To my eye, it looks like baseline docking is decreased in the distal AZ (200-600 nm). Not sure I understand this because unc-68 and unc-2 have similar decreases in mini rate. This (and the preceding comment #6) could indicate that the spatial pattern of SV priming/docking is altered by the CaV2 and RYR mutations

The reviewer notes a visible decrease in docking in the distal zone in the mutants. We compared the number of docked vesicles in distal regions (165-594 nm). As suggested, there are fewer docked vesicles in the unstimulated condition of RyR(-) and CaV1(-) mutants compared to the wild type (Figure 3 supplement C). However, the decrease was not observed in the CaV1(-) CaV2(-) double mutant, which is not consistent with a requirement for CaV1 or the RyR for docking at lateral sites, and makes interpretations muddied. It is nevertheless possible that calcium from inner stores is required to facilitate docking – possibly by activating UNC-13S. We have added Figure 3 supplement C to provide a statistical analysis of docking at lateral sites, but have not considered any speculations in the text concerning potential roles of calcium in docking.

Figure 3 supplement: Figure legend updated.

8) Are lateral SV fusions (33-166 nm) significantly decreased in unc-2 mutants (Figure 3E)? This would suggest that CaV2 does promote lateral SV fusions, potentially by activating RYR channels.

The reviewer is correct; there are statistically fewer fusions in the intermediate zone (33nm to 165nm). We defined this domain to minimize calcium from CaV2, which is likely predominant at 33nm. The number of docked vesicles remaining after stimulation is significantly higher in CaV2(-) and in RyR(-) mutants compared to the wild type (Figure 3 supplement B). The number of synaptic vesicles after stimulation in these mutants is the same (Welch’s T-test P=0.8908), indicating that CaV2 and RyR have similar fusion defects in this zone. As proposed by the reviewer, these data support the conclusion that CaV2 (as well as CaV1) activates the ryanodine receptor.

p9-10, “In the intermediate active zone (33-165 nm) there was no change in the baseline docking of any mutant (Figure 3 Supplement B). However, we observed identical fusion defects in the intermediate zone in the CaV2(-) mutant and RyR(-) mutant, which suggests CaV2 activates RyR to release calcium from internal stores. Activation of RyR by CaV2 is consistent with the multiquantal release mediated by these channels (Figure 2E-G).

9) Are distal SV fusions (198-400 nm) exaggerated in unc-2 mutants (Figure 3E)? If so, how do you explain this?

The reviewer asks if there are more fusions in the most distal zone in the CaV2 mutants. The implication is that there may be compensatory fusion at the distal sites. We compared the number of docked vesicles present in the most distal active zone (165-594nm) in the unstimulated and stimulated conditions. The number of docked vesicles before stimulation in the CaV2(-) mutant was no different than the wild-type counterpart, however after stimulation significantly fewer docked vesicles remained in the CaV2(-) mutant. Moreover, the fusion probability in the most distal active zone is significantly increased in CaV2 mutants (wild type: 68.6%, *unc-2*(-): 72.1%; Chi-Squared P=0.013).

p10. In the distal active zone (165-594 nm), the probability of vesicle fusion was reduced in CaV1(-) mutants, RyR(-) mutants, and CaV2(-) CaV1(-) double mutants, but slightly increased in CaV2(-) mutants (Figure 3 Supplement D). An increase in fusion at distal sites in CaV2 mutants, might be due to compensatory effects in the expression or organization of CaV1 and RyR.“

10) Please provide a more complete description of the super-res imaging methods (Figures 4-8). I'm not sure that I fully understand these data. The images shown are apparently derived from single-molecule localizations. So, the macroscopic puncta shown in these figures result from superimposing many localizations? If so, please indicate how many localizations are shown for each image. Can you estimate the absolute number of molecules present in each punctum? This would be interesting, especially for CaV2, CaV1, RYR, and UNC-13. Is there any substructure evident in these signals, e.g. nanoclusters of CaV2, CaV1, or RYR. If so, it would be very interesting to provide details about these nanoclusters (size, spatial frequency, number of molecules in each).

The reviewer is asking two questions: First, are the intensely staining clusters of channels saturated at their centers? Second, can we estimate the number of channels at synapses?

Method: See improved description under Methods.

Display. The reviewer’s understanding of the data is correct. The images are plots of localizations from thousands of frames, each point is an (x,y) coordinate from a single blink. These data are collected using a ground-state depletion (GSD) protocol, so that a labelled protein can be stimulated and fluoresce multiples rounds, rather than in PALM mode in which fluors are bleached and thus removed from the sample. GSD leads to an improved signal-tonoise ratio, better spatial resolution, and less undercounting which are all chronic problems in PALM microscopy, but quantitation of proteins is more complicated using GSD. In our data sets, the mean precision of all blinks is 40 nm. For plotting localizations, blinks that had less than an 80 nm precision were discarded, and each blink is plotted using an 80 nm arbitrary point size to reflect this confidence interval. The plotted puncta could arise from blinks from different channels in close proximity, or multiple blinks from a single channel. The intensity and thus saturation of signal is a readout of the number of localizations overlapping in an area. We have included a plot of a synapse which uses translucent dots to illustrate both the center of the blink and confidence interval in the localization, rather than arbitrary dots to represent calcium channels (Reviewer Figure 1). This plot may help the reviewer better understand the discussion of superresolution data, but these plots are visually deceptive because the eye is drawn to the larger, less accurate, localizations. To ascertain protein abundance, rather than just localization, we need to squeeze every last blink from the tag to perform a robust Poisson statistical analysis of the data. However, in the 3-protein localization experiments in this manuscript, we needed to both retrieve a significant number of localizations for each fluor, and also preserve the remaining probes for subsequent localizations. These limitations are described below.

**Author response image 1. sa2fig1:** Calcium channel localizations visualized by sphere sized by radial precision. It should be noted that large dot size corresponds to less-certain localizations rather than the size of the calcium channel cluster. The most accurate localizations are small dots. CaV1 yellow. CaV2 magenta.

Quantification: Methods for quantification of proteins using suicide enzymes could be limited by the efficiency of the enzymatic reaction with the enzyme. We tested labeling efficiency by sequential labeling using different dyes, and found that our protocol saturated both the HALO and SNAP enzymes.

We performed a Fourier ring analysis of our data and observe a 40 nm resolution, meaning that we can distinguish two points if they are at least 40 nm apart.

Limits imposed by multiprotein localizations: In our experiments three proteins were localized at the same synapses. A disadvantage of multiprotein localization is that recording conditions must be optimized to retrieve localizations from all 3-colors but imaged sequentially. There are three reasons that our protocol could fail to localize all proteins in the sample.

(1) Undersampling. Imaging was performed to prevent bleaching subsequent dyes, that is, limited excitation was used to preserve the sample. Thus, some fluors remained dark during recording.

(2) Ground-state depletion. Dyes were likely destroyed during the intense stimulation required to get all fluors into the dark state.

(3) Bleaching. Subsequent fluors, particularly red-shifted fluors are bleached from a prior excitation laser, despite gentle laser settings.

Estimates: Despite these concerns we placed upper and lower limits for the number of channels as described below.

CaV1 calculations. CaV1-SNAP was coupled to JF549cp ('cell permeable'). CaV1 channels are dispersed in the synapse but also often form small clusters, spatially distinct from CaV2 clusters (Figure 10A). To estimate the number of CaV1 channels, we optimized imaging conditions to maximize localizations from only CaV1 by imaging them first, and estimated their number using three approaches.

(1) Total blinks. We assumed each blink is a channel (total CaV1 blinks Ᾱ = 196 ± 37 channels) (Figure 10B-C). Total blinks could possibly represent an undercount. Because the process of placing fluors in the dark state requires intense illumination until all fluorescence is in the dark state, some fluors may have been driven to a bleached, and thus irretrievable, condition. On the other hand, fluors blink multiple times using a ground-state depletion protocol. Since total blinks assumes every channel blinks only once, this method will represent an overcount of the true number of channels. Because we observe multiple blinks from single channels (see next method), we think the value of 200 channels likely represents the maximum number of channels.

(2) Mean blinks per channel (~60 nm). Spatial information was used to determine how many blinks belong to the same channel – singlet blinks were assigned as a channel, and two or more blinks within 60 nm of each other were assumed to arise from a single channel based on a typical confidence interval (our plots exhibit a 40 nm resolution by Fourier ring analysis, to be conservative we use a 60 nm arbitrary criterion given that we only include blinks with better than 80 nm localization precision by Cramér-Rao lower bound; for context, a calcium channel with subunits is ~20 nm in diameter). A Poisson distribution was fit to the number of blinks per channel (m = 2.7 blinks per channel). Dividing the total number of blinks in synapses by the mean number of blinks per channel produces an estimate of 79 ± 10 CaV1 channels per synapse (Figure 10B,C).

(3) Photon flux. Every blink has a photon count. The mode number of photons emitted in a blink was determined from *all blinks* at analyzed synapses (Mo = 575 photons / blink; Ᾱ = 765 ± 11 photons/ blink) (Figure 10D,E). Blinks were also clustered into single channels (as described above) and the number of *photons per ‘channel’* was measured. Channels emitted a mean of 1996 ± 489 photons because some channels blinked multiple times. The mode was 550 photons per channel (Figure 10D-E). The mode photon count for channels is similar to the mode photon count of single blinks indicating that most channels blink once.

The total number of photons emitted by a synapse for the whole imaging session was divided by the mean number of photons per channel to estimate CaV1 channel count: (75 ± 19 CaV1 channels)

Methods associating overlapping blinks can produce an undercount of the true number of channels due to limited resolution; overlapping confidence intervals from adjacent channels will erroneously “chain” multiple channels into a single channel. Methods with a greater than 10nm precision will be needed in the future to accurately count channels.

Nevertheless, these last two estimates agree quite well. We now report the mean of these values: 77 ±15 CaV1 channels per synapse.

CaV2 calculations. CaV2-HALO was labelled with JF646. CaV2 channels are tightly localized into ~250nm diameter clusters coincident with the dense projection. Each synapse contains just one density, but each density has several hundred CaV2 blinks. The number of CaV2 channels in individual boutons was estimated with the following approaches.

(1) Total blinks. CaV2 exhibited a mean 569 ± 54 blinks / synapse. Again, as described above total blinks is an overestimate of the number of channels at a synapse.

(2) Mean blinks or photon flux per channel. CaV2 blinks exhibited a mode of 900 photons. However, we were unable to associate blinks with a single channel; CaV2 localizations saturate the dense projection. The cluster appears as a uniform mass in our images, because our resolution and precision are not adequate to spatially resolve single channels in densely labeled domains. To circumvent this limitation, the maximum number of channels in the density was calculated.

(3) Cluster diameter – upper bound. We can estimate an upper bound for the number of channels by mathematically fitting circles within a larger circle (20 nm channels in a 250nm cluster) yields a maximum capacity of 120 channels at the dense projection.

(4) Minimal overlap -- lower bound. The CaV2 clusters appears continuous without cavities, meaning all localizations overlap with neighboring channels. A mathematical fit of 60 nm circles within a 250 nm cluster results in at least 12 channels. Each channel would need to blink 50 times in 1 minute. Organic dye blinking photophysics are not well characterized. However, this rate is approximately 10-fold higher than rates experimentally determined, as described below, suggesting there are many more channels.

(5) Blinking rate. The blinking rate of dyes was used to estimate channel count. Although blinking rates of rhodamine dyes have not been determined, cyanine dyes in oxygen scavenging buffer can blink about 6 times in one minute depending on the imaging conditions (Helmerich et al. 2022). We observe 569 blinks on average from the CaV2 cluster in a one minute imaging interval. If rhodamine dyes are similar to cyanine dyes, then 101±16 CaV2s are present at a single synaptic bouton. We think 101 CaV2 channels is the best estimate given our limited knowledge of dye photophysics.

(6) Calcium sensitivity. Although not fully characterized, we and others suspect calcium causes more rapid blinking of the dye (based on our observation of occasional propagation of a wave of blinks). If this assumption is correct then CaV2-HALO labelled with JF646 may blink more often than solitary tagged proteins because they are tightly localized at the dense projection in close proximity to other potentially open calcium channels, which would cause an overestimate of channels at the dense projection.

We conclude based on the blinking rate and limits to the size of the dense projection that there are approximately 100 CaV2s at each bouton.

We have added these points to the Discussion, and have been more explicit in the Methods regarding sample preparation for superresolution imaging. We have also included plots from CaV1 CaV2 and CaV1 ELKS RyR imaging experiments to illustrate photon counting from individual synapses (Figure 10F,G).

p14, "The number of calcium channels per synapse can be estimated from the single molecule localization data (see Methods for details). CaV1 channels are dispersed in the synapse but often also form small clusters (Figure 10A, F, G). The number of CaV1 channels was calculated from the mean number of blinks per channel, as well as from the total photon flux, converging on 77 ± 15 CaV1 channels per synapse (Figure 10B-E).

The ryanodine receptor images were suffused with high background fluorescence, and photon flux was not a reliable measure. Using the mean blinks per channel produces an estimate of 29 ± 4 RYR channels per synapse (Figure 10G).

CaV2 channels are tightly localized to the dense projection (Figure 10F). In our images the cluster appears as a solid mass; the overlap in localization precision made it impossible to assign blinks or photon flux to individual channels in the cluster. However, assuming the blinking rate of rhodamine dyes is similar to cyanine dyes (Helmerich et al., 2022), the frequency of blinking indicates that the cluster contains 101 ±16 CaV2 channels.

The ~100 CaV2 channels per synapse derived from our single molecule localization data is much higher than the ~35 CaV2.1 channels determined by immunogold labelling synapses in the mouse central nervous system (Holderith et al., 2012; Kusch et al., 2018). Nevertheless, the density of calcium channels at *C. elegans* neuromuscular junctions (91 CaV2 per µm^2^) is similar to mammalian synapses (100-400 CaV2.1 channels per µm^2^). “

11) Do you have data showing that the tagged UNC-2, EGL-19, UNC-68, and UNC-13 alleles have no effect on function?

To illustrate the effects of tagging the calcium channels, we have included behavioral data of the multiply tagged strains in Figure 4 supplement E-I. All of the strains are superficially normal in morphology and locomotion. In addition, we performed a detailed behavioral analysis. The strains are normal for crawling speed, forward run distance, and backward run distance. However, almost all the multiply tagged strains exhibit an increase in reversals, and animals with tagged RyR or UNC-13all exhibit an increase in distance travelled. We conclude that multiply tagged strains have minor defects for specific behaviors, but gross locomotion is not affected.

p10, "The strains used for 3-color imaging exhibited normal morphology and appear to move like wild-type animals, suggesting the tagged proteins are functional. Analysis of specific movements indicated that most locomotory responses are normal; however, the frequency of reversals was increased in all multiply tagged strains (Figure 4 Supplement E-I).”

We have added tagged strain locomotion plots to Figure 4 supplement and updated the figure legend.